

# Radar-Based Quantitative Precipitation Estimation for the Identification of Debris-Flow Occurrence over Earthquake affected Region in Sichuan, China

Zhao Shi[1,2,3,4,5], Fangqiang Wei[1,2,3], Chandrasekar Venkatachalam[4]

[1]Key Laboratory of Mountain Hazards and Earth Surface Process, Chengdu, 610041, China
[2]Institute of Mountain Hazards and Environment, Chinese Academy of Sciences, Chengdu, 610041, China
[3]University of Chinese Academy of Sciences, Beijing, 100049, China
[4]Colorado State University, Fort Collins, 80523, USA
[5]Chengdu University of Information and Technology, Chengdu, 610225, China

*Correspondence to*: Zhao Shi (shi_zhao@foxmail.com)

**Abstract.** Both of Ms8.0 Wenchuan earthquake on May 12, 2008 and Ms7.0 Lushan earthquake on April 20, 2013 occurred in Sichuan Province of China. In the earthquake affected mountainous area, a large amount of loose material caused a high occurrence of debris flow during the rainy season. In order to evaluate the rainfall Intensity–Duration (I-D) threshold of the debris flow in the earthquake-affected area, and for filling up the observational gaps caused by the relatively scarce and low altitude deployment of rain gauges in this area, raw data from two S-band China New Generation Doppler weather radar (CINRAD) were captured for six rainfall events which triggered 519 debris flows between 2012 and 2014. Due to the challenges of radar quantitative precipitation estimation (QPE) over mountainous area, a series of improving measures are considered including the hybrid scan mode, the vertical reflectivity profile (VPR) correction, the mosaic of reflectivity, a merged rainfall-reflectivity(R-Z) relationship for convective and stratiform rainfall and rainfall bias adjustment with Kalman filter (KF). For validating rainfall accumulation over complex terrains, the study areas are divided into two kinds of regions by the height threshold of 1.5 km from the ground. Three kinds of radar rainfall estimates are compared with rain gauge measurements. It is observed that the normalized mean bias (NMB) is decreased by 39% and the fitted linear ratio between radar and rain gauge observation reaches at 0.98. Furthermore, the radar-based I-D threshold derived by the Frequentist method is $I = 10.1D^{-0.52}$, and it's also found that the I-D threshold is underestimated by uncorrected raw radar data. In order to verify the impacts on observations due to spatial variation, I-D thresholds are identified from the nearest rain gauge observations and radar observations at the rain gauge locations. It is found that both kinds of observations have similar I-D threshold and likewise underestimate I-D thresholds owing to under shooting at the core of convective rainfall. It is indicated that improvement of spatial resolution and measuring accuracy of radar observation will lead to the improvement of identifying debris flow occurrence, especially for events triggered by the small-scale strong rainfall process in the study area.



# 1 Introduction

Rainfall-induced debris flow is a kind of ubiquitous natural hazard for the mountain area with complex terrain. It is a geomorphic movement process which scour the sediment from steep areas into alluvial fans. The formation of rainfall-induced debris flow is generally related to three main factors, including the gravitational potential energy, abundant loose materials and meteorological events (Guzzetti et al.2008). The gravitational potential energy relatively remains stable for a long period of time. The loose materials are normally made up of sand, unsorted silt, cobbles, gravel, boulders and woody debris (Qiang W. et al., 2015). High magnitude level earthquake events can generate abundant loose solid material from co-seismic rock falls and landslides, and deposited in gullies (Shieh et al. 2009). During the rainy season, the occurrence of Debris flow after an earthquake becomes more frequent (Yu et al. 2014; Xiaojun G., et al 2016). Both the Ms 8.0 Wenchuan earthquake on May 12, 2008 and the Ms 7.0 Lushan earthquake on April 20, 2013 occurred in Sichuan province of China and have changed the formation conditions for debris flow. A large number of debris flow occurred from 2008 to 2014 and caused lots of casualties and extensive property damage.

Early Warning System (EWS) for rainfall-induced landslide and debris flow are widely implemented in many parts of the world (e.g. Segoni et al., 2015; Michele C. et al.,2015; Baum and Godt, 2010).The performance of Early Warning System (EWS) highly rely on the updating of precipitation thresholds (Rosi et al., 2015). Furthermore, considering the material condition of forming debris flow is vastly changed at earthquake affected region (Tang et al., 2009, 2011, 2012), it is necessary to revaluate the precipitation threshold. The model of rainfall Intensity-Duration is widely used to represent the precipitation thresholds of triggering landslides and debris flow (Aleotti, 2004; Guzzetti et al., 2007). Some literatures concluded that the I–D relationships for some of the regions were severely affected by the Wenchuan Earthquake (e.g. Su et al., 2011, 2012; Zhou et al., 2012, 2013; Cui et al., 2013; Guo et al., 2015). However, most of these I-D relationships are derived from rain gauge observation. This is a common technical way to estimate the I-D thresholds of debris flows using rainfall observation from the nearest rain gauge. However, the uncertainty of intensity-duration thresholds from rain gauge observations could not be ignored. This is related to two critical limitations which probably lead to underestimation of observation of strong convective events occurring at high altitude area. The first limitation is the relatively sparse network density of rain gauges in the mountainous region (Francesco M. et al., 2014), the other one is the altitude of gauge deployments which is at low elevation for sustainability. The same limitations of rain gauge observation also exist in the mountainous regions of Sichuan province. The technique of microwave remote sensing has become a necessary way for observing rainfall events in complex terrain. The radar-based quantitative precipitation estimation (QPE) has been demonstrated useful for the study of debris flows, as its unique advantage of high spatial and temporal resolution. Radar observation offer the unique merit of estimating rainfall over the actual debris flow location (David N. et al., 2004; Chiang and Chang, 2009, Francesco M. et al., 2014; M. Berenguer, 2015). However, there are many challenges when Radar-based QPE in the mountainous area is applied in the study of debris flow. Commonly, keeping the elevation angle close to the ground and estimating the sample cut at the same height is a basic requirement for radar QPE to represent the actual rainfall distribution on the ground. The radar beam blocked by the mountain





is a serious problem for the low angle observation. The radar beam angle have to be elevated to avoid the blockage. However, doing this introduces another problem which is rainfall distribution at higher heights is different from that of at the surface and it also varies largely according to the precipitation type (Jian Z.et al., 2012). Errors due to radar system calibration and uncertainty in hydrometeor's DSD (Drop Size Distribution) also decrease the accuracy of rainfall estimates. Furthermore,

debris flow triggering events are often related to high precipitation gradients of storms which occur for a short duration and have small scale (Nikolopoulos et al., 2014). Considering these, raw S-band radar reflectivity data are used to estimate rainfall, and assess the impact of estimation errors on the identification of I-D threshold over the study area.

The main aim of this study is to merge the radar QPE, thereby improving its estimation over complex terrain and to assess the impact of rainfall estimate accuracy on the identification of I-D threshold over the study area. To do that, a series of accuracy-

improving measures have been adopted including a hybrid scan mode, the vertical reflectivity profile (VPR) correction, the mosaic of reflectivity, a combination of rainfall-reflectivity(R-Z) relationship for convective and stratiform rainfall and rainfall bias adjustment with Kalman filter (KF). Three radar rainfall estimation scenes are evaluated with the rain gauge observations for six debris-flow triggering rainfall event to validate the accuracy of radar estimate. I-D thresholds are identified from 519 rainfall-induced debris flow events with the frequentist method (Brunetti et al., 2010; Peruccacci et al., 2012). Another aim of

this study is to understand the impact on the I-D identification due to spatial variability of rainfall observation. Rain gauge observations closest to the debris flow within 10 km and radar observations at the rain gauge locations are used to get the I-D relationship.

## 2. Study domain and data

The study area is located at Sichuan Province in southwest China which consists of 16 administrative districts and counties.

The area of study is about 38,000 km$^2$ and occupies nearly 8% of the land area of Sichuan Province (see Figure 1). This area was strongly affected by the Ms8.0 Wenchuan earthquake which occurred in 12$^{th}$ May, 2008 and the Ms7.0 Lushan earthquake which occurred in 20$^{th}$ April, 2013. In the following years, debris flow happened frequently. During the period from 2012 to 2014, the debris flow occurring in this area accounted for 58.3% of the annual debris flows events which occurred in the whole province. The area is in the transitional zone of the Qinghai-Tibet Plateau to the Sichuan Basin. Terrain changes steeply and

the average altitude above sea level (a.s.l) for this area is between 500 meters and 6 kilometres. Bedrock outcrops in this region are weak and prone to physical weathering. Loose quaternary sediments are distributed in the form of river terraces and alluvial fans (Tang et al., 2011).

The climate type of the study area is humid subtropical. The monthly precipitation distribution is commonly affected by the plateau monsoon, the East Asian monsoon and complex terrain. The mean annual rainfall over the central and southern parts

of this region varies from 1200 to 1800 mm, and sometimes even reaches 2500 mm (Xie et al., 2009).The mean annual rainfall





over the western part of this area is less than 800 mm. The north and southwestern areas of this region are in the transition zone from hot dry to humid climates, with mean annual precipitation ranging between 800 and 1200 mm.

The area is monitored by two well-maintained S-band Doppler weather radars (see Figure 1). One is deployed in Chengdu city with an altitude of 596m above the sea level and the other one is deployed at Mianyang city with the height of 557m above the sea level. Both of the radar systems have same performance specifications which can be seen in table 1. The system provides radar rainfall estimates at a radial range resolution of 300 m and an angular resolution of 1 degree. There is a rain gauge network consisting of 551 gauges equipped at the meteorological surface station in the study areas. The number of rain gauges seems to be a lot, but most of them are deployed at the valleys. The density of rain gauges is severely scarce at the high altitude of the mountain, resulting in observation gaps where the debris flow initially takes place. The average altitude above sea level of those rain gauges is far lower than 3km.

Six debris-flow triggering rainfall events which occurred in the area of study between 2012 and 2014 are analysed. Those events happened at the most severe earthquake affected region during rainy season and triggered a total of 519 debris flow that caused casualties and extensive property damage. Table 2 summarizes the characteristics of the rainfall events. Three events occurred in August, two events occurred in July and one occurred in June. These events are deemed to be representative of the debris flow-triggering precipitation in the region during the rainy season. The event duration time and maximum rainfall accumulation are also retrieved by the rain gauge closest to debris flow location and radar observations. The identification of rainfall event was determined by an interval of at least 24 hours, the rain rate is less than 0.1 mm/h (Guzzetti et al., 2008; F. Marra., 2014). Table 2 indicates that the durations and rainfall accumulations identified by gauge and radar are different owing to the precipitation type and density of rain gauges. The identification differences of event No.1, 2, and 6 between gauge and radar are not so large like event No. 3, 4, and 5. From the figure 2 of radar-estimated rainfall accumulation for the six rainfall events (the improving measures described below are applied in the figure 2), it can be seen that the precipitation of event No.3, 4 and 5 is dominated by convective and the strong core of rainfall region is located at the high altitude area where rain gauge is relatively scarce. A few of debris flow occurred at the long range, approaching radar detection edges, while the rainfall measured there was low. This may be caused by the decreasing resolution at long radial range. In following section, the distance and height are considered as an evaluation factor to assess the radar-based rainfall estimate.

## 3. Methods

### 3.1 Radar accumulated rainfall estimation methods

S-band weather radar has a unique advantage of being unaffected by attenuation, as it is subjected to Rayleigh scattering for almost all hydrometeors. However, in complex terrain conditions, S-band radar observations still face serious challenges. The main problem comes from ground clutter and severe beam blockage, resulting in inaccurate estimates of radar rainfall. A number of signal processing techniques have been developed to detect and remove clutter and anomalous propagation (AP), including fuzzy logic, ground echo maps, Gaussian Model adaptive processing (GMAP) filter, etc. (e.g. Berenguer et al., 2006;



Cuong m. n. and V. Chandrasekar, 2013; Harrison et al., 2000). For the radar data used in this study, ground clutter is filtered with the GMAP algorithm configured in Vaisala Sigmet digital processor. Furthermore, in order to overcome the beam blockage and ensure the rainfall estimation accuracy, radar data are corrected concerning the following issues: (i) Beam shielding and hybrid scan, (ii) Vertical profile of reflectivity, (iii) Mosaic of hybrid scan reflectivity, (iv) Combination of

reflectivity rainfall relationship, (v) Rainfall bias adjustment.

*Beam shielding and hybrid scan.* The mode of Hybrid scan is used to form the initial reflectivity field for rainfall estimate, by keeping the radar main beam away from the blockage of the complex terrain. (Jian Z. et al., 2012). In the study area, the grids with 0.36 km$^2$ resolution on the ground are aligned with radar bins of each elevation angle. The blockage coefficients of the low elevation angles at 0.5°, 1.5° and 2.4° are calculated according to the Digital Elevation Model(DEM), earth

curvature, antenna pattern and the wave propagation model (Pellarin et al., 2002;Krajewski et al., 2006). The blockage ratio distribution of two S-band radar can be seen from figure 3. There are almost no topographical shielding in the near field within the distance of 50 km from each radar. The main factor considered in the hybrid scan within 50 km is to meet the estimated rainfall from the same vertical height as much as possible. Thus the area within 20 km from radar is assigned with the elevation angle of 3.4°, the area from radar between 20 and 35 km is assigned with the elevation angle of 2.4°, the area from radar

between 35 and 50 km is assigned with the elevation angle of 1.5°. It is assigned with the elevation angle of 0.5° by default, if there is no blockage over 50km distance from the radar. The terrain transforms from plain land to mountainous region over about 70 km westward away from each radar. At this region the altitude rises sharply, and elevation angle of 0.5° is totally obscured. Therefore, the lowest angle at which the blockage ratio does not surpass 0.5 is assigned to the aligned grid. Meanwhile, the blockage ratio is correspondingly used to compensate the energy loss of reflectivity. The final adaptive-terrain

hybrid scan maps are combined as shown in figure 3 (d) and (h). It can be seen that most of the study area are covered by the 1.5° and 2.4° of radar scan.

*Vertical Profile of Reflectivity (VPR).* Due to the hybrid scan, the radar elevation angle is raised resulting in majority of the observed reflectivity coming from the upper levels of precipitation profiles. This is quite different from the actual reflectivity

on the ground. It is necessary to account for the reflectivity correction at the ground level. This study adopts the AVPR method to adjust the reflectivity (Jian Z., et al., 2012). The processing steps applied in this study include: (i) To discriminate convection precipitation from stratiform based on the composite reflectivity>50dBz or VIL >6.5 kg/m$^2$, (ii) The parameterization of VPR is carried out to generate bright band top, peak, bottom heights and piecewise linear slope $S_1$, $S_2$, and $S_3$(see Figure 4). (iii) Reflectivity observed is adjusted based on the parameterized VPR to piecewise extrapolate the corresponding reflectivity at

the ground.

Figure 4 shows a sample scatter plot of the vertical reflectivity profiles from 11: 30 to 12: 30 on July 21, 2012. It can be seen directly that the changing characteristic of vertical reflectivity rely on the temperature, air dynamic, practical size, etc. Considering altitude is one the critical factor affecting the atmosphere physics parameters and the performance of VPR. The





areas of study are classified as two types: region type I and II with the condition of the height from the ground (≤1.5 km for region type I and >1.5 km for region type II) and the distance from the radar (≤100 km for region type I and >100 km for region II). Figure 5 shows the identification result for both radars. Apart from the VPR adjustment, these two kinds of regions are respectively assessed during the validation of radar QPE, in order to understand the actual impact of distance and height of radar observations

5   on the rainfall estimation.

***Mosaic of hybrid scan reflectivity.*** Both of two S-band radar have common coverage areas where reflectivities data should be mosaicked to construct a large-scale sensing for rainfall events. Taking the distance and altitude as weighing parameters, the mosaic formula is define below:

$$Z_M = \frac{\sum_i w_i \times k_i \times Z_i}{\sum_i w_i \times k_i}$$

(1)

and

$$w_i = exp\left(-\frac{d_i^2}{L^2}\right)$$

(2)

$$k_i = exp\left(-\frac{h_i^2}{H^2}\right)$$

(3)

Here, $Z_M$ represents the mosaicked hybrid scan reflectivity, $Z_i$ is the single radar hybrid scan reflectvity, i is the radar index, $w$ is weighing component for the horizontal weighting, and $k$ is weighing component for the vertical weighting. The variable d is the distance between the analysis grid and the radar, and h is the height above the ground of the single radar hybrid scan. The parameters L and H are respectively scale factors of the two weighting functions.

***Combination of rainfall relationship,*** Rainfall rates are calculated from radar reflectivity by a power law empirical

relationship which is called *Z–R* relationship (Austin 1987; Rosenfeld et al. 1993), and theoretically, the Z–R relationships should be adjusted when the drop size distributions (DSD) change over the rainfall duration. However, it is still a challenge to obtain fine spatial distribution of DSD with change of time over complex terrains. This study adopts the two widely verified Z-R relationships defined as: Z = 300R^1.4 for convective precipitation (Fulton et al. 1998) and Z = 200R^1.6 for stratiform (Marshall et al. 1955), and the rainfall type is identified during VPR processing.

***Rainfall bias adjustment.*** The errors of R-Z relationship mainly come from raindrop size distribution (DSD) variation, radar calibration errors etc. (Berne and Krajewski, 2013), so the rainfall bias change over time. The mean field bias correction is a method to calculate the ratio of the means of radar estimate and the rain gauge observation (Anagnostou and Krajewski, 1999; Chumchean et al., 2003; Yoo and Yoon, 2010). In this study, the bias is calculated based on hourly radar rainfall accumulation and rain gauge accumulated observation. It's defined as:




$$BIAS = \frac{\frac{1}{N}\sum_i^N \mathrm{r}_i}{\frac{1}{N}\sum_i^N \mathrm{g}_i} \qquad (4)$$

where $BIAS$ is mean rainfall bias in one hour, g is one hour accumulated rainfall of rain gauge, i is rain gauge index, r is the radar-based one hour accumulated rainfall over the i-th rain gaugee and N is the total number of rain gauges. As described above, the density of rain gauge deployment over the mountainous area is relatively scarce. Therefore the precipitation measured by individual gauges at high and low altitudes may lead to overestimation and underestimation respectively. Therefore, the Kalman filter is adopted to alleviate the measurements noise of the bias (Ahnert et al., 1986;Chumchean et al., 2006; Jungho K. and Chulsang Y. 2014).

The basic steps of Kalman filter in this study include:

Step 1 State estimate prediction:

$$BIAS_P(n) = BIAS_{KF}(n-1) \qquad (5)$$

where:$BIAS_P$ represents the bias prediction, $BIAS_{KF}$ represents the bias estimate update, n is discrete time.

Step 2 State estimate error covariance prediction:

$$P_P(n) = F^2 \times P_{KF}(n-1) + Q \qquad (6)$$

where: $P_P$ represents the bias estimate error covariance prediction. $P_{KF}$ represents the bias estimate error covariance update. Q represents covariance function of the system error.

Step 3 Calculating Kalman gain

$$G(n) = P_P(n) \times (P_P(n) + S)^{-1} \qquad (7)$$

where: $G$ represents the Kalman gain. S represents covariance function of the measurement error.

Step 4 Updating State estimate

$$BIAS_{KF}(n) = BIAS_P(n) + G(n) \times \left[BIAS_m(n) - BIAS_P(n)\right] \qquad (8)$$

where:$BIAS_m$ represents the bias measurement.

Step 5 Updating estimate error covariance

$$P_{KF}(n) = (1 - G(n)) \times P_P(n) \qquad (9)$$

It is assumed that the variation of the real bias between each hour is tiny, while the measurements of bias for each hour vary largely, so initial conditions of KF are that: Q=0.1, S=10, $BIAS_{KF}(1) = BIAS_m(1)$, $P_{KF}(1) = 1$.

## 3.2 Intensity-Duration threshold identification methods

Rainfall thresholds for the possible initiation of debris flows are identified according to the I–D power law relationship (Guzzettiet al., 2007). , it's defined as bellows:

$$I = \alpha D^{-\beta} \qquad (10)$$





Calculating the event duration (D) and the average intensity (I) requires the start and end times of the rainfall event. The duration and intensity of each debris flow can be directly identified with the time-sequential radar rainfall estimate .These times are determined by an interval of at least 24 hours, rain rates of less than 0.1 mm h$^{-1}$ (Guzzetti et al., 2008; F. Marra., 2014), or correspondingly radar reflectivity of less than 10 dBz to separate two consecutive rainfall events. The parameters

of $a$ and $\beta$ are estimated with the Frequentist method (Brunetti et al., 2010).

In order to illustrate the impacts of radar rainfall estimate on I-D threshold, basic procedures of the frequentist method are applied to radar rainfall accumulation and are described below:

(i) Radar-identified rainfall durations and average intensities are log transformed as log(I) , log(D). Both of them are fitted by least square method to form a linear equation as $\log(I) = \log(\alpha_{50}) - \beta \log(D)$ , where $\alpha_{50}$ , $\beta$ here account for

nearly 50% occurrence probability of debris flow.

(ii) For each debris flow, the difference $\delta(D)$ between the actual rainfall average intensity $\log[I(D)]$ and the corresponding fitted intensity value $\log[I_f(D)]$ is calculated, $\delta(D) = \log[I(D)] - \log[I_f(D)]$.

(iii) The probability density function (PDF) of the of $\delta(D)$ distribution is determined through Kernel Density Estimation and furthermore fitted with a Gaussian function, which is defined as:

$$f(\delta) = a \times exp\left(-\frac{(\delta - b)^2}{2c^2}\right)$$ **(11)**

where $a > 0$, $c > 0$, and $a, b, c \in \mathbb{R}$.

(iv) The threshold for expected minimum exceedance probability ($P_{mep}$) is determined by PDF function, as

$$\int_{-\infty}^{\delta_{mep}} f(\delta)d\delta = P_{mep}$$ **(12)**

where $\delta_{mep}$ is the intercept parameters. $\delta_{mep}$ can be resolved through Equ.(12) for given $P_{mep}$ , then the $\alpha_{mep}$

corresponding to the $P_{mep}$ is calculated as

$$\alpha_{mep} = \alpha_{50} exp(\delta_{mep})$$ **(13)**

Finally, $\alpha_{mop}$ and $\beta$ are best fitted parameters for exceedance probabilities $P_{mep}$.

The minimum exceedance probability is set to 5% for this study.

## 4. Result and discussion

The accuracy and robustness of the I-D threshold of the debris flow are determined by the accuracy of rainfall observation and positioning. Therefore, a series of processing including hybrid scan, VPR correction, a combined R-Z relationship, and mean bias adjustment is performed on six rainfall event to improve the accuracy of radar-based accumulated rainfall. In order to evaluate the overall performance and verify the impact on I-D threshold due to rainfall accumulation accuracy, the assessment





was performed towards three scenes of radar-based estimates: scene I, the estimate from raw data of hybrid scan without VPR and bias adjustment; scene II, the estimates with VPR adjustment after scene I; scene III, the estimates with rainfall bias correction after scene II. According to rainfall estimate evaluation, I-D thresholds are derived from those scenes and also assessed concerning accuracy and spatial resolution.

5 **4.1 Assessment of rainfall estimation accuracy**

The accuracy of the radar-based event rainfall accumulation is assessed with the rain gauge observation. In order to perform evaluation, a set of criterions is calculated including normalized standard error (NSE), normalized mean bias (NMB) and correlation coefficient (CORR), defined as below:

$$NSE = \frac{\frac{1}{N}\sum_i^N |r_i - g_i|}{\frac{1}{N}\sum_i^N g_i} \times 100\% \qquad (14)$$

$$NMB = \frac{\frac{1}{N}\sum_i^N (r_i - g_i)}{\frac{1}{N}\sum_i^N g_i} \times 100\% \qquad (15)$$

$$CORR = \frac{\sum_i^N (g_i - \overline{g})(r_i - \overline{r})}{\sqrt{\sum_i^N (g_i - \overline{g})^2}\sqrt{\sum_i^N (r_i - \overline{r})^2}} \qquad (16)$$

where NMB and NSE are in percent, CORR is dimensionless, $r_i$ and $g_i$ represent the rainfall accumulation from radar and gauge, $N$ is the total sampling number. The statistical criterions comparisons between radar-rain gauge and the three radar estimate scenes are shown in table 3, and the scatter plot of radar-based estimates and rain gauge rainfall observations are

15 shown in figure 6. The comparison for scene I indicates: The NSE, NMB and CORR of the whole study areas are 50.7%, -41.1% and 0.78 respectively. The radar-based rainfall is underestimated, the linear ratio of rainfall observation between radar and gauge is 0.51, as shown in figure 6(a). The reason of underestimation is the systematic bias and uncertainty of reflectivity on the ground. From the comparison of two type regions, it can be observed, the NSE, NMB and CORR of region type I are relatively better than region type II. It is revealed that improving measure are needed for the hybrid scan estimate.

The comparison for scene II indicates: The NSE, NMB and CORR for the study areas are 46.1%, -18.6% and 0.80 respectively. It is an improvement compared with the scene I. The radar-based rainfall is also underestimated through the VPT adjustment, and the linear ratio of rainfall observation between radar and gauge is 0.76, as shown in figure 6(b). This means rainfall biases still exist in the estimate. The NSE and CORR of region type I are also slightly better than region type II.

The comparison for scene III indicates: The NSE, NMB and CORR of the whole study areas are 44.0%, 1.91% and 0.84

respectively. The linear ratio of rainfall observation between radar and gauge is 0.98, as shown in figure 6(c), and this means the consistency between rainfall and radar observation is achieved through the Kalman filter-based bias correction. Figure 7 shows the average and covariance of bias estimation by Kalman filter and mean field bias method for six rainfall event. The



CORR and NSE improvement also verify the effectivity of Kalman-filter for radar QPE in mountainous areas, Kalman filtering makes the whole rainfall event estimate free from large significant overestimation or underestimation.

Scene III provides the optimum rainfall estimation for this study. In the following, all of the three scenes are used to assess the impact of QPE accuracy on I-D relationship identification.

## 4.2 Intensity-duration thresholds based on radar QPE

The radar rainfall estimates with high spatial resolution can retrieve rainfall duration and average intensity for each rainfall-induced debris flow, so an abundant of sample data are captured to induce the I-D relationship. Scatter distribution of event duration-intensity for the three radar estimated scenes are shown in figure 8, Comparisons of scatter distribution between each other's indicate that the average rainfall intensity and duration are incrementally increased when applying the improving measures. The PDF estimations reveal that the number of positive difference $\delta(D)$ is more than the number of negative difference. This can be accounted for storm triggering which is relatively dominant. The parameters of Gaussian function are summarized in table 4. The parameter $a$ is incrementally decreasing. When applying the improving measures, parameter c has the opposite changing trend and parameter b is randomly changes around the small range of zero.

The I-D threshold derived from the scene III is $I = 10.1D^{-0.52}$ . It is higher than the other two I-D thresholds derived from scene I and scene II, owing to application of accuracy improving measuring.

### 4.3 comparison with intensity–duration thresholds from rain gauge observations

In order to analyze the impact of the spatial sampling variability on identification of I-D threshold for radar estimate and rain gauge observation, I-D threshold are derived from rain gauge closest to the debris flow and radar estimate at the corresponding the co-location of rain gauge(Francesco M. et al.,2014). There are some same predefined conditions for comparison: (1) duration times are identified separately by two kinds of sensors, rainfall duration time is required to be more than 1 hour and minimum mean rainfall rate is 0.1 mm/h. (2) the maximum distance from debris flow location is less than 10km.(3) identification of I-D threshold calculated from frequentist methods with exceedance probabilities of 0.5%.

Firstly, the event rainfall accumulation are compared between rain gauge observations closest to the location of debris flows and radar estimates at the location of the corresponding rain gauge. The scatter plot of rain gauge and radar estimate is shown in figure 9. The corresponding metrics are calculated. The CORR is 0.88, NMB is 17.07%, NSE is 28.32% and linear ratio is 1.13, indicating that rainfall observations from rain gauge closest to debris flows location and radar estimate at co-location have the tendency of consistency.

The I-D threshold are derived from rain gauge and radar estimate. Scatter plots of I-D pairs are shown in Figure 10,The I-D threshold estimated from rain gauge is $I = 5.1D^{-0.42}$, The other I-D threshold estimated from radar is $I = 5.8D^{-0.41}$.Both of I-D thresholds seem little lower than $I = 10.1D^{-0.52}$, since the scarce gauge network didn't capture the strong core of rainfall





which triggered the debris flow. It is interesting to note that both I-D thresholds of radar and rain gauge are very similar, although there are some measurement errors between them as shown in figure 9.

## 4.4 Comparison with previous results

The I–D threshold for the study regions is compared with other global, regional thresholds in the literature. It can be seen from figure 11 that the thresholds obtained in this work (red in Figure 11) fall in the range of other I-D thresholds. The results were also compared with the rainfall thresholds previously proposed in the Wenchuan Earthquake area (Tang et al., 2012; Wei Z. and Tang, 2014; Xiaojun Tang, 2016). Our result lies at the middle range of them. The difference comes from the database we used, the radar data which is used to fill the observation gap of rain gauges, and the identification method of I-D threshold were also different due to a different exceedance probability. The I–D threshold of this study is crossed checked with that proposed in the Chi-chi Earthquake affected area in Taiwan (Chen et al., 2005), mainly owing to the climatic differences like storm occurrence duration and intensity. The result nearly overlapped with the one proposed in Adige area of Italy(Francesco M. et al.,2014). The I-D threshold is lower than the cases for Japan (Jibson et al., 1989) and for the world (Caine et al., 1980), but higher than the world (Guzzetti et al., 2008).

## Summary

The main purpose of this paper is to evaluate the debris flow occurrence thresholds of the rainfall intensity-duration in the earthquake-affected areas of Sichuan province over the rainy seasons from 2012 to 2014. The paper calculates the Intensity-Duration threshold from radar-based rainfall estimate, which is different from the common method of using rain gauge observation. Radar observations have high spatial resolutions sensitive to convective precipitation, which is a critical issue for rain gauge observation owing to its scarcity and low altitude deployment over mountain areas. However, the accuracy of radar-based QPE over complex areas is affected by the terrain and remains a challenge for hydrological application. The following works were done to draw the conclusions.

(a) There are two S-band Doppler radars covering the study area. Radar observations for six rainfall events were processed with a series of mountain-oriented QPE algorithms including: a terrain-adapted hybrid scan, VPR correction, the reflectivity mosaic, the combination of R-Z relationships, and rainfall bias correction. Three types of estimation from radar are performed and compared with rain gauge observations to validate the accuracy. The results show that: The combination of the whole correction procedures reduces the bias to 1.91% and the NSE to 44%, meanwhile improves the correlation coefficient to 0.84 and the linear ratio to 0.98.

(b) Intensity Duration rainfall thresholds for the triggering debris flow are calculated with a frequentist approach. The I-D threshold of $I = 10.1D^{-0.52}$ is derived from the kalman filter corrected radar estimates. The accumulated rainfall is lower than rain gauge observations and the derived I-D is also underestimated. The hybrid scan, VPR correction and combination of R-Z relationship are strongly required.





(c) The I-D deduced from rain gauge observations closest to the occurrence of debris flow is highly similar to the one deduced from the radar estimates at the same location as rain gauge, which are $I = 5.1D^{-0.42}$ and $I = 5.8D^{-0.41}$ respectively. These I-D thresholds are underestimated owing to the rainfall spatial variation and the incontinuous sampling effect.

Finally, it is clear that radar-based rainfall estimate and threshold supplement the monitoring gap of EWS where rain gauge is scarce. A better understanding of relationship between rainfall and debris flow initiation for earthquake affected area can be enhanced by improving the spatiotemporal resolution and low elevation angle coverage of radar observation, especially for monitoring the convective storm occurring at the mountains.

**Acknowledgements.** This research was supported by the National Natural Science Foundation of China (No. 41505031), China Scholarship Council (No.201508515021), the Science and Technology Support Project of Sichuan Province (No. 2015SZ0214), Scientific Research Funding of CUIT (No. J201603), China Meteorological Bureau Meteorological Sounding Engineering Technology Research Centre Funding. We appreciate the weather bureau of Sichuan Province for the data services.

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



Table 1. Characteristics of S-band doppler weather radar.

| Items | Value |
|---|---|
| Wavelength | 10.4 cm |
| Polarized mode | horizontal |
| Antenna gain | 45 |
| First sidelobe level | -29 dBc |
| Peak transmitted power | 750kW |
| Pulse width | 1 µs |
| Noise figure | 4dB |
| Dynamic range | 90dB |
| Range resolution | 300m |
| Volume scanning elevation | 0.5°, 1.5°, 2.4°, 3.4°, 4.3°, 6.0°, 9.5°, 14.5°, 19.5° |
| Altitude above sea level of radar location | 595m for Chengdu site<br>557m for Mianyang site |



Table 2. Characteristics of the rainfall events

| Event No. | Date | Number of triggered debris flows | Event Duration by rain gauge (h) | Event Duration by radar (h) | Max. rainfall accumulation by rain gauge(mm) | Max. rainfall accumulation by radar(mm) |
|---|---|---|---|---|---|---|
| 1 | Jul. 9, 2012 | 9 | 12 | 11 | 17.5 | 29.6 |
| 2 | Jul. 21, 2012 | 9 | 10 | 12 | 29.3 | 23.6 |
| 3 | Aug.17–18, 2012 | 200 | 7 | 49 | 19.2 | 195.8 |
| 4 | Jun. 19, 2013 | 15 | 5 | 12 | 55.3 | 101.8 |
| 5 | Jul. 8-12, 2013 | 261 | 55 | 73 | 562.2 | 416.9 |
| 6 | Jul. 10-12, 2014 | 25 | 20 | 21 | 28.5 | 17.8 |





Table 3. The comparison of radar rainguage for each estimate scene

| Criterions | Scene I (Hybrid scan) | | | Scene II (VPR) | | | Scene III(Bias adjustment) | | |
|---|---|---|---|---|---|---|---|---|---|
| | Region Type I | Region Type II | All study Region | Region Type I | Region Type II | All study Region | Region Type I | Region Type II | All study Region |
| NSE (%) | 46.4 | 50 | 50.7 | 45.8 | 49.0 | 46.1 | 43.5 | 47.2 | 44.0 |
| NMB (%) | -40.9 | -42.8 | -41.1 | -17.1 | -21.2 | -18.6 | 1.7 | 10.8 | 1.91 |
| CORR | 0.80 | 0.77 | 0.78 | 0.82 | 0.77 | 0.80 | 0.85 | 0.82 | 0.84 |





Table 4. The parameters of Gaussian fitting which are used by frequentist method to account for I-D threshold

| Parameters Of Gaussian fitting | Scene I | Scene II | Scene III |
| --- | --- | --- | --- |
| a | 3.144 | 2.55 | 2.22 |
| b | 0.011 | 0.003 | -0.003 |
| c | 0.1273 | 0.1578 | 0.1868 |


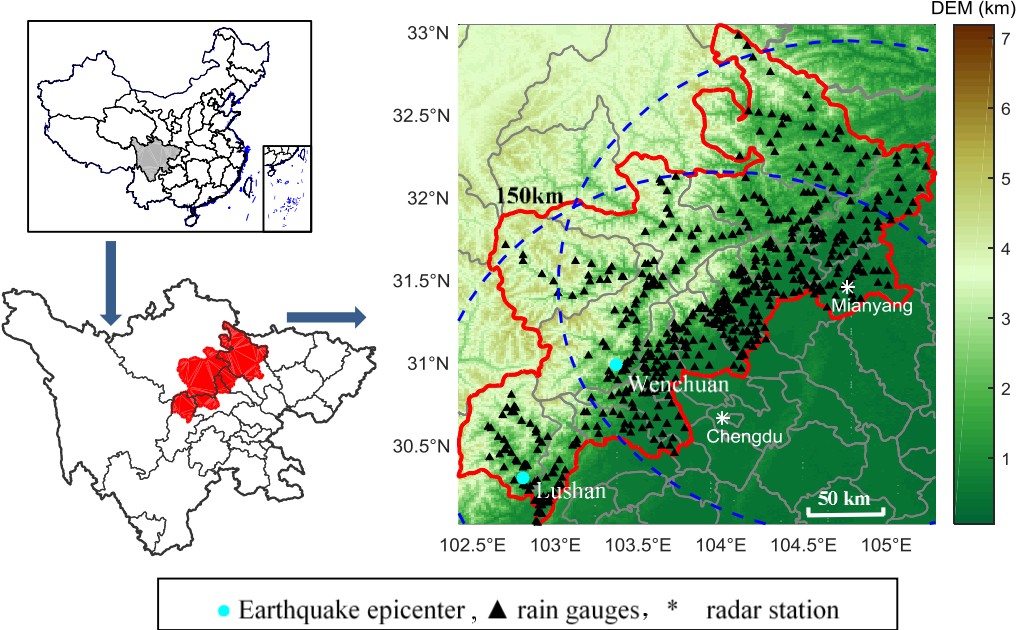

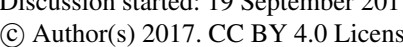



**Figure 1. Location and Topography of the study area. Asterisk markers show the location of Chengdu and Mianyang S-band weather radar which respectively monitor the study area within 150 km (dash black circle) from radar location. Rain gauges in the study area are marked with black triangle and mostly deployed at the valley. The two blue circle dots are the epicentre of Ms8.0 Wenchuan earthquake in12ᵗʰ May , 2008 and Ms7.0 Lushan earthquake in20ᵗʰ April, 2013.**





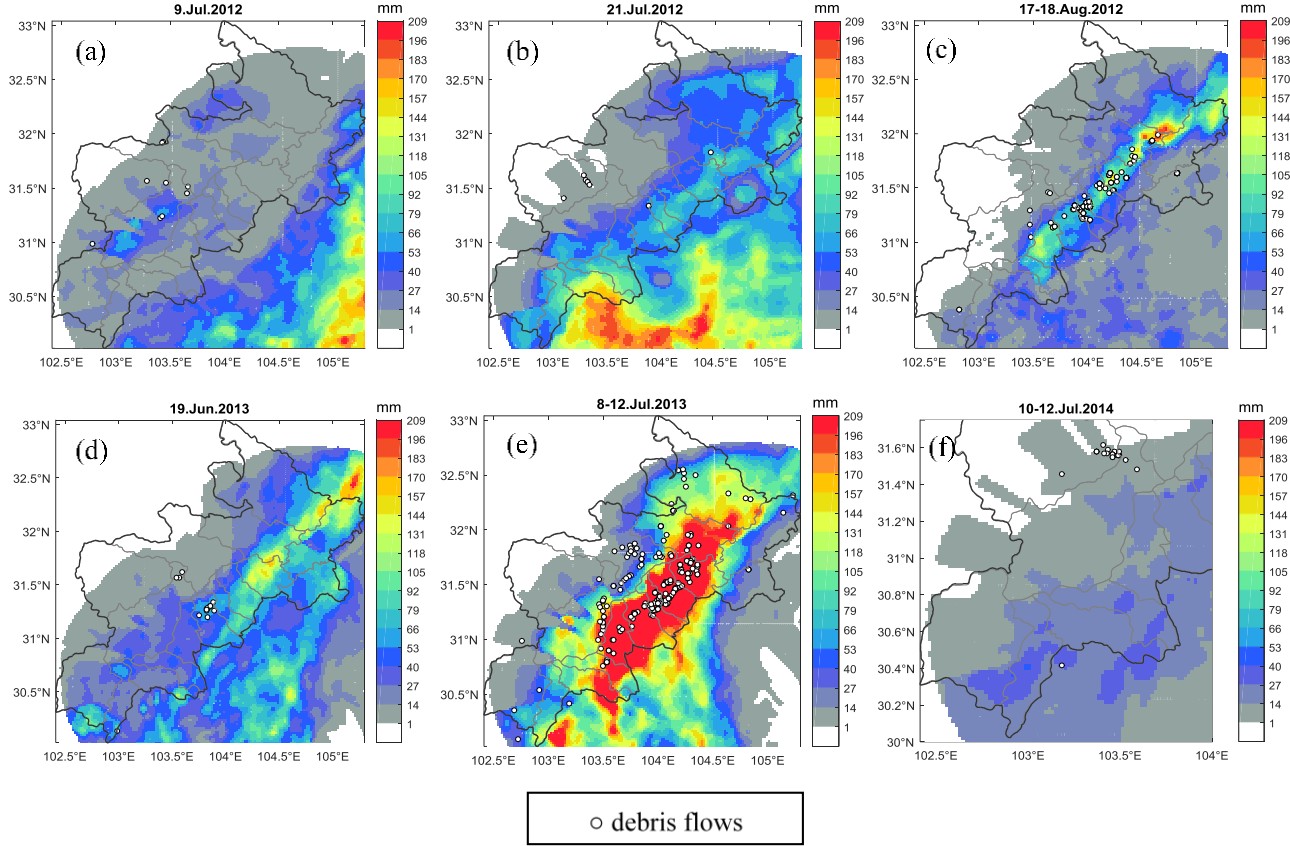

**Figure 2. Images of radar-estimated rainfall accumulation for the six rainfall events (a–f). dotted circles represent the location of triggered debris flows. Events are showed in chronological order: (a) 9th July 2012; (b)21th July 2012; (c) 17-18th August 2012, (d)19th June 2013; (e)8-12th July 2013; (f)10th,12th July 2014.**





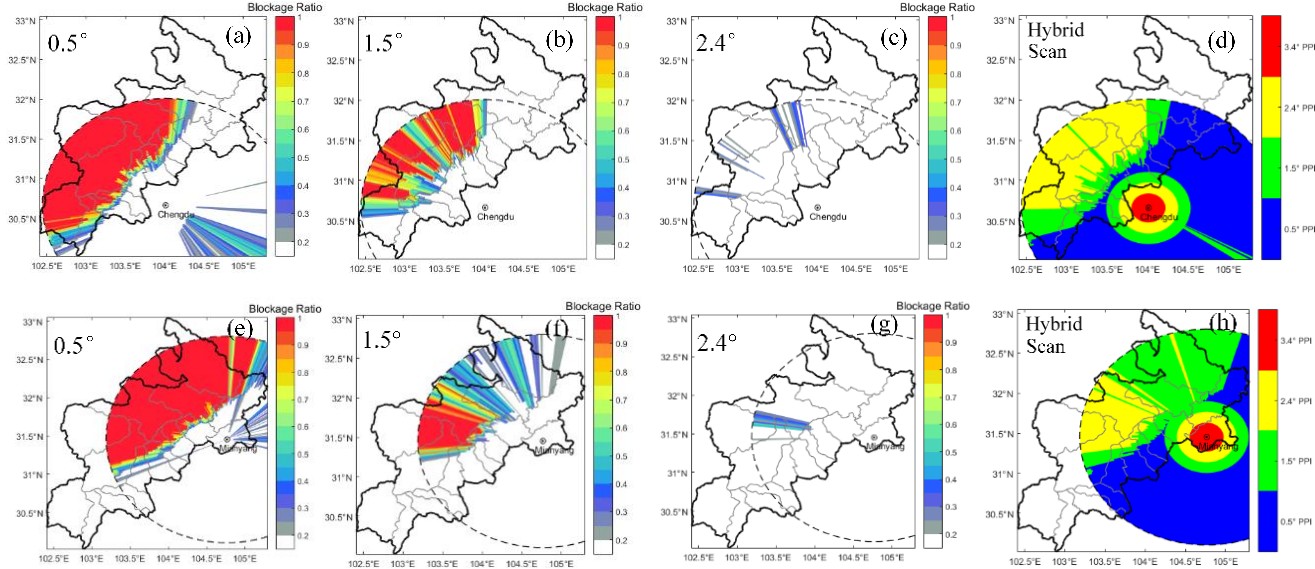

**Figure 3. Blockage ratio of beam shielding for the radar main lobe beam and hybrid scan map. (a)-(c) represent the blockage ratio of Chengdu radar at the elevations of 0.5º, 1.5 º and 2.4 º respectively. (e)-(f) represent the blockage ratio of Mianyang radar at the elevations of 0.5º, 1.5 º and 2.4 º respectively. Hybrid scan maps for Chengdu and Mianyang are merged under the condition of blockage raito is lower than 0.5.**





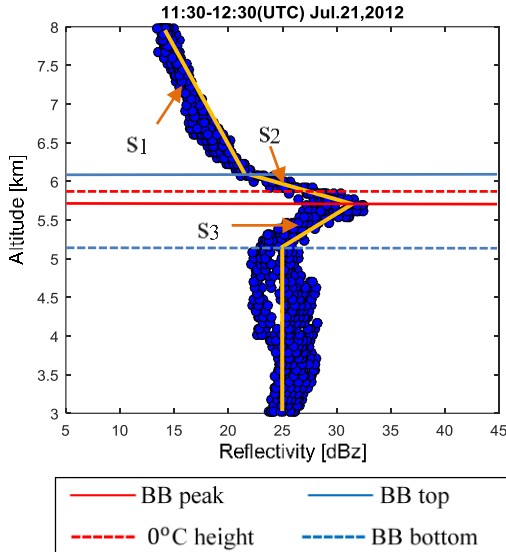

**Figure 4. A real sample of VPR model processed in the study on Jul.21,2012. The Blue circle represents azimuthal mean of reflectivity over one hour. The orange line represents the idealized VPR with piecewise linear slope α,β, and γ. The horizontal blue lines is the bright band(BB) top and dashed blue lines is BB bottom. The solid red line and dashed red lines are BB peak and the 0°C height, respectively.**


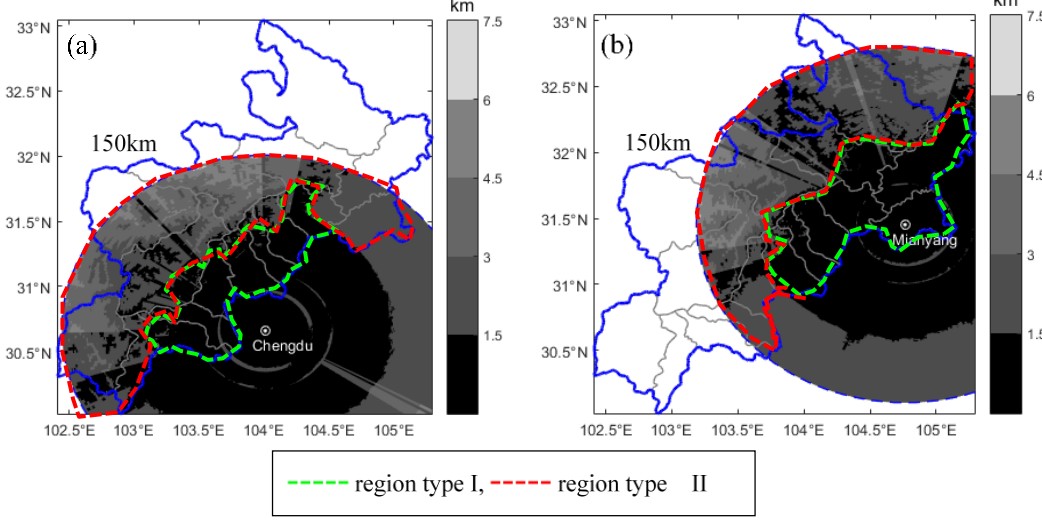

**Figure 5.** The height from the ground of hybrid scan for two S-band radar (a) radar located at Chengdu (b) radar located at Mianyang . The regions surrounded by green dash lines meet the condition of that the height from the ground is 1.5 km below and the distance from radar is inner 100 km and is recognized as region type I. The regions surrounded by the red dash lines represents the area under the opposite condition and is recognized as region type II.



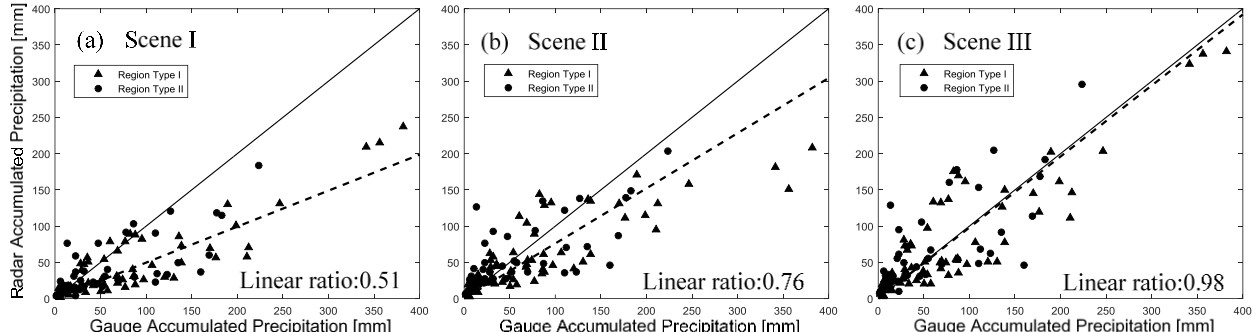

**Figure 6. Scatter plots of radar and rain gauge event-rainfall accumulations. (a) Scene1: radar estimate from hybrid scan. (b) Scene 2: radar estimate from hybrid scan and VPR. (c) Scene 3: radar estimate through the hybrid scan, VPR and bias correction.**





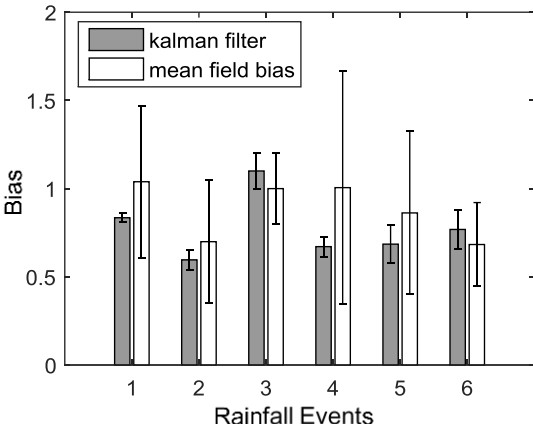

Figure 7. The average and covariance of bias estimation by Kalman filter and mean field bias method for six rainfall event.





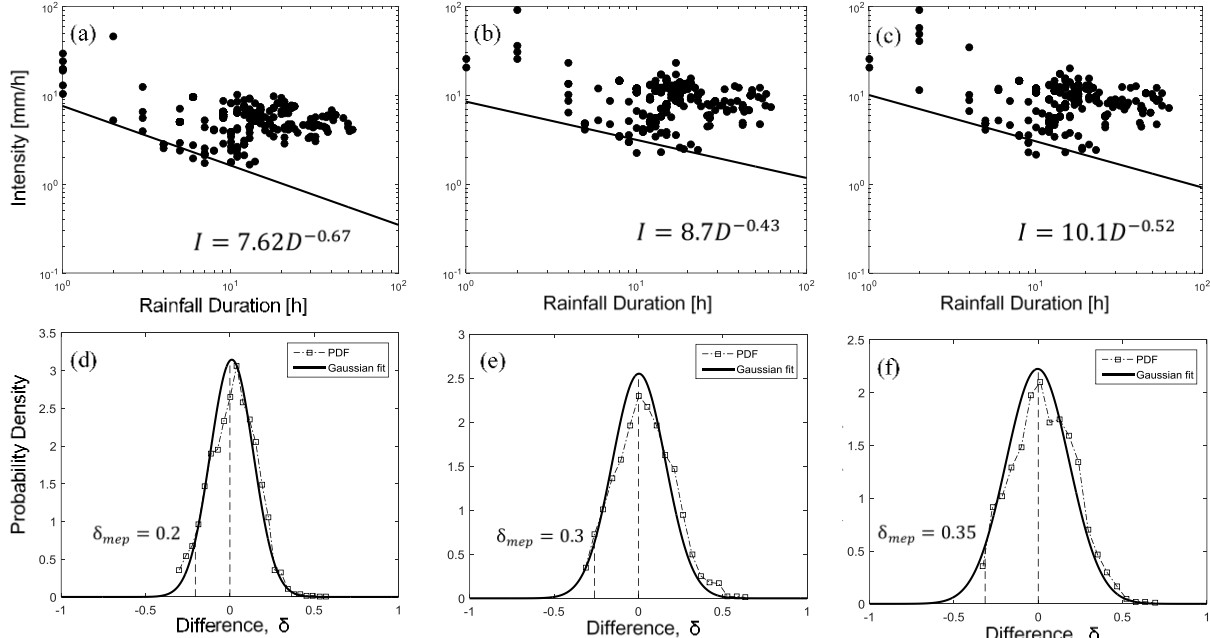

**Figure 8. Scatter plots of radar and rain gauge event-rainfall accumulation and possibility density function. (a), (b), (c) are the scatter plot of scene I, II, III respectively.(d),(e) and (f) are the Gaussian fitted PDF of scene I, II, III respectively.**

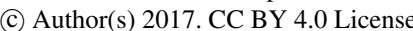


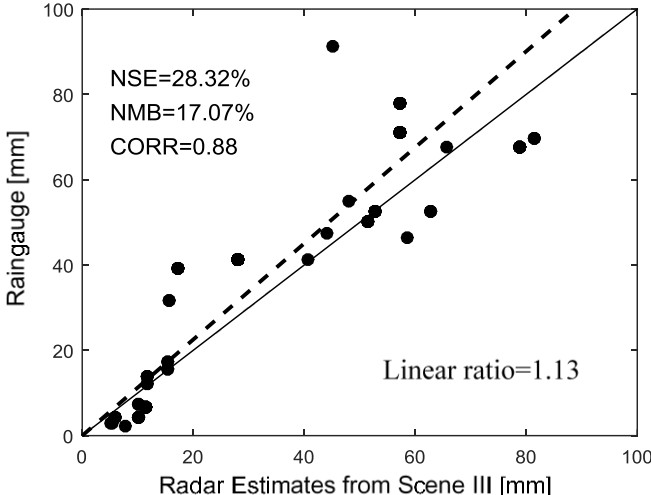

**Figure 9. Event-rainfall scatter plots of rain gauges closest to debris flow locations and radar-based estimate from scene III over the same location of rain gauge.**





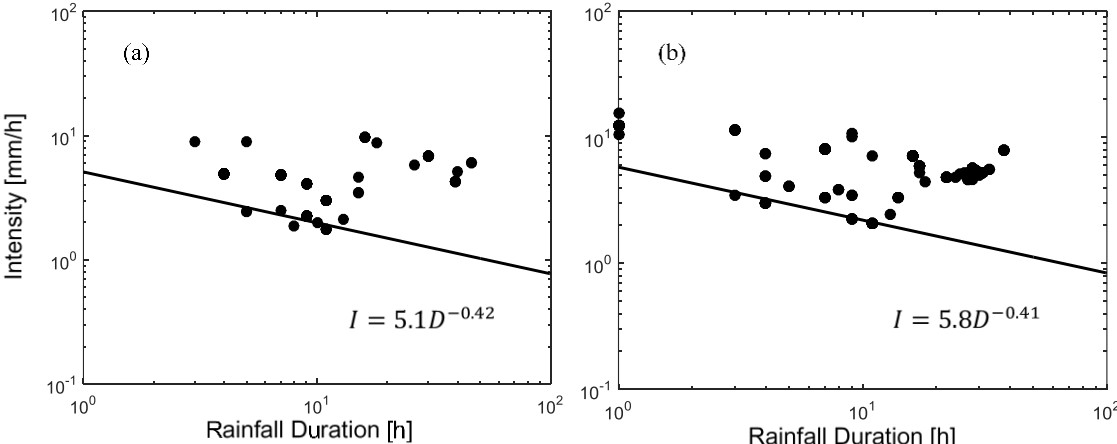

**Figure 10. Intensity–duration thresholds (black line) derived from (a) rain gauges closest to debris flow locations and (b) radar rainfall estimation at the same location of the rain gauges closest to the debris flow.**





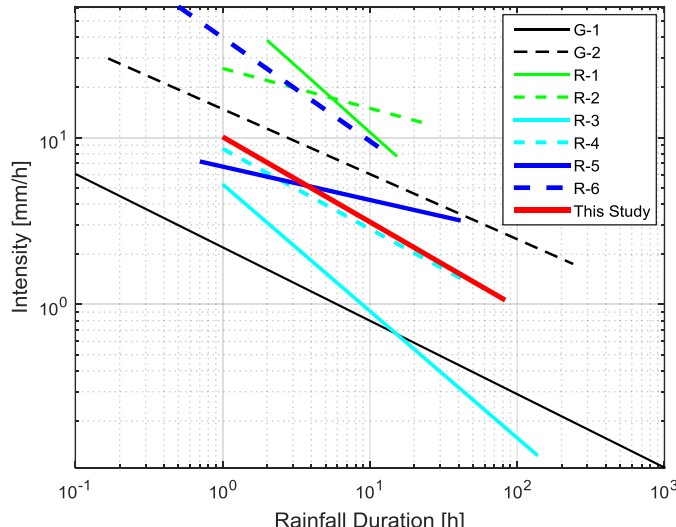

**Figure 11. I–D thresholds determined for this study (red line) and those of various other studies. G = global, R = region. G-1: Guzzetti et al. (2008); G-2: Caine (1980); R-1: Wenchuan earthquake area, Wei and Tang(2014); R-2: Qingping, a region in Wenchuan earthquake area, Tang et al. (2012); R-3: Wenchuan earthquake area, Xiaojun G. et al.(2016); R-4: Italy, Francesco M. et al.(2014); R-5 Central Taiwan, Jan and Chen(2005); R-6 Japan, Jibson (1989).**