# Peer review of "Radar-Based Quantitative Precipitation Estimation for the Identification of Debris-Flow Occurrence over Earthquake affected Region in Sichuan, China"

_Natural Hazards and Earth System Sciences, 2017_

## Referee Comment (RC1) · Anonymous Referee #1 · 18 Oct 2017

Zhao Shi et al. MS No.: nhess-2017-308

General comments The paper deals with debris-flow occurrence thresholds in terms of rainfall intensity-duration by using radar data in an earthquake-affected area (Sichuan, China). The paper addresses technical questions within the scope of NHESS and is conform to international standards; scientific methods are clearly outlined. Authors try to address the definition of the rainfall-field by the use of radar to better evaluate the relation between rainfall and debris-flow. But, as described in the technical comments, to overcome some objective limitations of the methodology used, further investigations

would be needed to properly take into account the susceptibility of the territory. The title is clear; the abstract is pertinent, easy to understand and resumes well the contents of the paper. Mathematical formulas, symbols and abbreviations are correctly defined. Technical language is precise and understandable.

An I-D threshold curve is proposed on the basis of six rain events that caused 512 debris-flows between 2012 and 2014. Based on rain data estimates obtained from 2 radars, the authors use various techniques for estimating rainfall from reflectivity, by also considering the rainfall detected by rain gauges for comparative analysis and correction of the bias. As shown, the use of radar data provides a better estimate of the precipitations responsible of landslides triggering. In one case (cf. Table 2), the amount of rain measured by radar is 10 times greater than that detected by the rain gauge network (Event # 3). Before evaluating the I-D threshold, the authors describe some techniques to detect the best estimate of radar precipitation, identifying the Kalman's filtering application as the best tool to reduce radar-rain gauge bias. Using the frequentist approach, the authors compute the relation $I = \alpha D\text{-}\beta$ which binds intensity and duration of precipitation for three estimates of rainfall fields. Scatter plots are shown in fig. 8 Technical Comments

The article is of interest as it performs a radar application on I-D threshold models, historically developed using only rain gauges. Although, I-D thresholds generally suffer from limitations that can affect practical applications. In fact, in case of convective rainfall events (responsible of most debris-flow and shallow-landslide events), the area affected by ground effects is usually small (normally few km2) and tends to reproduce a low I-D curve when only rain gauges are used. On the other hand, landslides occur in correspondence of highest rainfall intensities. Aiming at evaluating the precipitation field, authors try to solve the problem by also using radar data, and considering possible estimation errors. Another intrinsic lack of the method lies in the determination of the duration of the event, and the simplification of the average intensity or accumulated precipitation for the whole interval. Analyses of uniform rainfall distribution inside the intervals excludes soil characteristics that heavily affect the development of gravitational movements. In the case of stratiform rain events, with uniform intensity, landslides are more frequent in permeable areas; in case of convective rainfalls, landslides are more easily triggered in areas with low permeability. An additional limit of the model is the absence of discriminant analysis of susceptibility to shallow landslides in the study area. The adopted method for determining the threshold I-D, in fact, ignores lithology, geomorphology (e.g. slope), land use, and soil coverage. Consequently, it tends to lower the threshold as is results mostly controlled by the most fragile portions of the territory. Looking at the threshold I-D of Figure 8 for the same duration of precipitation, the corresponding rain intensity that triggers debris-flow varies with one order of magnitude. The paper, assessing thresholds I-D using radar rainfall field, contributes to improve the model by reducing limitations due to the use of only rain gauges, especially in the case of convective phenomena. Among possible improvements to the method, multiple thresholds (e.g. for class of slope, lithology, etc.) could be considered in addition to accurate rainfall radar estimation.

---

## Referee Comment (RC2) · Anonymous Referee #2 · 19 Oct 2017

General Comments: This manuscript investigates the debris flow occurrence using rainfall Intensity-Duration (I-D) information in the earthquake-affected areas in Sichuan, China. Two S-band Doppler radars are used to estimate rainfall in the study domain characterized by complex terrain. Therein, traditional Z-R relations are adopted with additional attention paid to vertical profile of reflectivity (VPR) correction and Kalman filter based bias correction. The I-D curves are then calculated using a frequentist approach with the radar derived rainfall products.

Overall, this topic well fits the scope of NHESS. The methodology used in the paper

is scientifically sound. The study shows several valuable scientific results, which can possibly be used for guidance of debris flow warnings. Nevertheless, some minor changes will improve paper clarity. Specific comments are listed below: 1. The writing needs to be improved. Please proofread the manuscript before submitting the revised version. Specific examples of mistakes are presented below: 1.1. Page 2, Line 15: highly rely-> highly relies 1.2. Page 3, Line 1: have to be -> has to be 1.3. Page 25, caption of Figure 7: event->events 1.4. Page 26, caption of Figure 8: possibility->probability

2. Reference formatting should be more consistent throughout the manuscript. There are numerous references that are not cited in correct form. For examples, Qiang W. et al., 2015; Xiaojun G. et al 2016; Michele C. et al., 2015; Francesco M. et al., 2014; and many others.

Additional Related Reference: Willie, D., H. Chen, and V. Chandrasekar, et al., 2017: Evaluation of Multisensor Quantitative Precipitation Estimation in Russian River Basin. Journal of Hydrologic Engineering, 22(5), E5016002, doi: 10.1061/(ASCE)HE.1943-5584.0001422.

3. Page 2, Line 14: Early Warning System Using acronym would be enough since it had been mentioned earlier.

4. Page 4, Line 5: performance specifications->system specifications

5. Page 5, Lines 14-16: The authors are using different tilt data trying to obtain rainfall estimates at the same vertical level. What is the purpose of doing this? Getting rainfall closer to surface might be more useful (provided that lowest level data is not blocked).

6. Page 6, Line 23: The rationale of using these two Z-R relations is insufficient. It is well known that Z-R is greatly dependent on local rainfall microphysics. A local DSD-based variability analysis would be helpful.

7. Page 15, Table 1: doppler->Doppler

8. Page 15, Table 1: The pulse width seems not matching with the range resolution. Please clarify.

---

## Referee Comment (RC3) · Anonymous Referee #3 · 26 Oct 2017

The manuscript presents an analysis of rainfall intensity-duration (I-D) thresholds used for the identification of debris-flow occurrence. Estimation of gauge and radar-based I-D threshold is carried out and compared. The work in this manuscript is very similar to the one carried out by Marra et al. 2014 thus from a methodological point of view there is no significant novelty. However, the authors carry out the analysis in a completely different region with different hydroclimatic characteristics and as such I consider the results to be complementary to what we already know from past studies. Therefore, I consider overall that the results reported in this work add to our knowledge and further

highlight the significance of using remote sensing observations for the estimation of debris flow triggering rainfall. I am including below a list of comments/suggestions that can hopefully help the authors to improve their manuscript.

1. Page 4, L11-25: This last paragraph should be placed in a different section (not the study area and data). You could have a dedicated section to discuss event characteristics. Also in that same paragraph, you mention info that relates to methodology (e.g. identification of individual rainfall event) that should be placed in the methodology section. 2. Provide a more detailed analysis of the comparison between radar rainfall estimates at DF (debris flow location) and closest-gauge estimates. For example, a graph showing relative error (y-axis) vs distance (between closest gauge and DF) would be informative. Using different colors per event (on such a graph) would also provide some more info. Lastly, it would be interesting to show that for both rainfall intensity and duration, since you are reporting differences in duration as well. Differences in duration, although important for building I-D thresholds, are not frequently explored. I believe adding some more info on this would strengthen the overall analysis. 3. Provide also quantification metrics for changes in I-D parameters ($\alpha$ and $\beta$). 4. A professional or native English speaker needs to carefully edit the manuscript for grammatical errors and inappropriate wording (e.g. p10L1 "effectivity" p10L7 "induce" etc) 5. P5L3: "ensure the rainfall estimation accuracy" is quite a strong statement. Please revise. 6. P5,L27: Define VIL 7. P5,L31-32: "It can be seen that ...rely on temperature, air dynamic ...". I don't think that these can be seen from Figure 4 alone. Please revise. 8. Do you have any justification for the choice of 1.5km as the height threshold for separating the two regions? 9. P7L24: "between each hour is tiny" perhaps should be "within each hour is negligible". Please check and revise accordingly 10. P7L25: "so initial conditions of KF are..." I don't believe that the exact numbers for Q, S etc are a result of the previously stated assumption. Revise accordingly. 11. Be consistent with the reporting of equations. Some are in text instead of being numbered as others. Also in P8,L9, you should write log[If(D)] instead of log(I). Revise also the sentence stating "$\beta$ here accounting for nearly 50% occurrence probability....". It is the

intercept, not the exponent that relates to the probability according to the frequentist approach you used. 12. P9,L1 and elsewhere: use "scenarios" instead of "scene" 13. Equations 14 and 15 have the same formula. Please revise 14. Define what do you mean by "linear ratio". 15. P10,L10-11: "The PDF estimations reveal that the number of positive difference $\delta(D)$ is more than number of negative difference". I am not sure what is the point you are trying to make here. Also, if you think that the distribution of residuals is asymmetric, then you should not fit a Gaussian distribution. This affects also the frequentist approach you followed. Please revise/clarify this point.
* * *

---

## Author Comment (AC4) · 6 Dec 2017

We thank the reviewer for many valuable and constructive suggestions. As reviewer's suggested, some analyses about relative errors of rainfall accumlation, rainfall intensity and duration versus distance were performed. We made revisions according to reviewer's helpful comments. The supplemented pdf includes our responses to the reviewer.

Please also note the supplement to this comment:

[Figure]

https://www.nat-hazards-earth-syst-sci-discuss.net/nhess-2017-308/nhess-2017-308-AC4-supplement.pdf

[Figure]

**Supplement:**

The manuscript presents an analysis of rainfall intensity-duration (I-D) thresholds used for the identification of debris-flow occurrence. Estimation of gauge and radar-based I- D threshold is carried out and compared. The work in this manuscript is very similar to the one carried out by Marra et al. 2014 thus from a methodological point of view there is no significant novelty. However, the authors carry out the analysis in a completely different region with different hydroclimatic characteristics and as such I consider the results to be complementary to what we already know from past studies. Therefore, I consider overall that the results reported in this work add to our knowledge and further highlight the significance of using remote sensing observations for the estimation of debris flow triggering rainfall. I am including below a list of comments/suggestions that can hopefully help the authors to improve their manuscript.

We thank the reviewer for many valuable and constructive suggestions. The followings are our responses to the reviewer.

1. Page 4, L11-25: This last paragraph should be placed in a different section (not the study area and data). You could have a dedicated section to discuss event characteristics. Also in that same paragraph, you mention info that relates to methodology (e.g. identification of individual rainfall event) that should be placed in the methodology section.

   Response: We thank the reviewer for this helpful suggestion. The contents of this paragraph are adjusted according to this comment. Those contents related to event characteristic and identification of individual rainfall event were moved to the section 4 in the revision.

2. Provide a more detailed analysis of the comparison between radar rainfall estimates at DF (debris flow location) and closest-gauge estimates. For example, a graph showing relative error (y-axis) vs distance (between closest gauge and DF) would be informative. Using different colors per event (on such a graph) would also provide some more info. Lastly, it would be interesting to show that for both rainfall intensity and duration, since you are reporting differences in duration as well. Differences in duration, although important for building I-D thresholds, are not frequently explored. I believe adding some more info on this would strengthen the overall analysis.

   Response: We thank the reviewer for the valuable and instructive suggestion. Based on the comment, we compared the relative errors versus the distance. Some metrics describing relative errors were introduced as below:

| Factors | Radar estimate at DF location versus rain gauge observation closest to DF location | Radar estimate at DF location versus Radar estimate at the position of closest rain gauge. |
|---|---|---|
| Accumulated Rainfall Relative Error (ARRE) | $ARRE_g(i) = \dfrac{\left\|R_{df}(i) - R_g(i)\right\|}{R_{df}(i)} \times 100\%$ | $ARRE_r(i) = \dfrac{\left\|R_{df}(i) - R_r(i)\right\|}{R_{df}(i)} \times 100\%$ |
| Duration Relative Error (DRE) | $DRE_g(i) = \dfrac{\left\|D_{df}(i) - D_g(i)\right\|}{D_{df}(i)} \times 100\%$ | $DRE_r(i) = \dfrac{\left\|D_{df}(i) - D_r(i)\right\|}{D_{df}(i)} \times 100\%$ |
| Rainfall Intensity Relative Error (RIRE) | $RIRE_g(i) = \dfrac{\left\|I_{df}(i) - I_g(i)\right\|}{I_{df}(i)} \times 100\%$ | $RIRE_r(i) = \dfrac{\left\|I_{df}(i) - I_r(i)\right\|}{I_{df}(i)} \times 100\%$ |

Note. R represents accumulated rainfall for debris flow event, D represents duration for rainfall event, I represents the mean intensity for rainfall event. The variables with subscript df, g and r

respectively represent the observation from radar at debris flow location, rain gauge closest to debris flow location, and radar at the position of closest rain gauge.

ARRE, DRE and RIRE are calculated for each debris flow event. Correspondingly, the distance from debris flow location to closest rain gauge is also calculated. Within 10 km range, scatterplots of the ARRE, DRE and RIRE versus distance are drawn in Figure 1.

[Figure]

Figure 1. Scatterplot of relative errors versus distance. Blue circle dot represent relative error between radar estimate at debris flow location and rain gauge observation closest to debris flow location. Red asterisk represent relative error between radar estimate at debris flow location and radar estimate at the position of closet rain gauge.(a) Accumulated Rainfall Relative Error, (b) Duration Relative Error, (c) Rainfall Intensity Relative Error.

Concentrating range of relative errors are summarized in Table 1.

Table 1. Concentrating range of relative errors for rainfall sensor closest to DF location

| Rainfall sensor closest to DF location | $\frac{\Delta I}{I}$ | $\frac{\Delta D}{D}$ |
|---|---|---|
| Rain gauge | [-0.33, 0.47] | [-0.15, 0.35] |
| radar | [-0.44, 0.36] | [-0.2, 0.30] |

We clarified those in the section 4.3 of revision.

3. Provide also quantification metrics for changes in I-D parameters (α and β).

Response: We thank the reviewer for this suggestion. The parameters of I-D threshold estimated from Scenario III was taken as a reference, the relative errors for α, β between various fitted I-D and the I-D fitted from scenario III were calculated, as shown in the following table. We clarified this in the section 4.3 of revision.

Table 2. Parameters of the identified ID thresholds and relative errors

| | $\alpha$ | $\frac{\alpha - \alpha_{S3}}{\alpha_{S3}} \times 100\%$ | $\beta$ | $\frac{\beta - \beta_{S3}}{\beta_{S3}} \times 100\%$ |
|---|---|---|---|---|
| Scenario I | 7.62 | -24.5 | 0.67 | 28.8 |
| Scenario II | 8.7 | -13.8 | 0.43 | -17.3 |
| Scenario III | 10.1 | 0.0 | 0.52 | 0.0 |
| Rain gauges | 5.1 | -49.5 | 0.42 | -19.2 |
| Radar rainfall estimate at gauge location | 5.8 | -42.6 | 0.41 | -21.2 |

\*$\alpha_{S3}$, $\beta_{S3}$ here equals to $\alpha$, $\beta$ estimated from Scenario III, respectively.

Table 1 indicates that improving the accuracy of rainfall estimate could decrease the relative errors of $\alpha$ and $\beta$, rainfall spatial uncertainty related to the rain gauge observation lead to underestimation of the I-D threshold for those rainfall events.

4. A professional or native English speaker needs to carefully edit the manuscript for grammatical errors and inappropriate wording (e.g. p10L1 "effectivity" p10L7 "induce" etc)

   Response: We thank the reviewer for this good suggestion. The whole manuscript were read and revised by native English speaker.

5. P5L3: "ensure the rainfall estimation accuracy" is quite a strong statement. Please revise.

   Response: We thank the reviewer for pointing this out. The aim of processing is to improve the rainfall estimation accuracy, the sentence is revised as "improve the rainfall estimation accuracy".(P5L15)

6. P5,L27: Define VIL

   Response: The definition of Vertically Integrated Liquid (VIL) is added in the manuscript as "To discriminate convection precipitation from stratiform based on the composite reflectivity>50dBz or VIL >6.5 kg/m2, where VIL is acronym of Vertically Integrated Liquid water content and it is an estimate of the total mass of precipitation in the clouds."(P6L8)

7. P5,L31-32: "It can be seen that . . .rely on temperature, air dynamic . . .". I don't think that these can be seen from Figure 4 alone. Please revise.

   Respone: We thank the reviewer for pointing this out. We clarified this in the revision as" Impacted by the temperature, air dynamic, particle size and phase are changed along the vertical falling. Figure 4 shows vertical profile of reflectivity varied approximately as three piecewise linear sections".(P6L14)

8. Do you have any justification for the choice of 1.5km as the height threshold for separating the two regions?

   Response: The reason of separating two region is the vertical variation of rainfall rate profile, especially for convective rainfall. Normally, the low scanning elevation PPI is used to estimate ground rainfall rate. Considering the limitation of scanning elevation, station height, and earth curvature, the height from flat terrain is nearly 1.5 km if the radial range is 100 km (normally under maximum detection range) away from radar, even for the lowest elevation of 0.5°. Concerning complex terrain of study area, the elevation almost has to be uplifted to avoid beam blockage when radial distance is over 100 km. Therefore, 1.5 km is set as height threshold for this study to discriminate where is closer to ground and where is higher from ground. The following figure briefly illustrate the radar beam locating height along the radial distance.

[Figure]

Figure 2. Radar beam locating height along the radial distance

9. P7L24: "between each hour is tiny" perhaps should be "within each hour is negligible". Please check and revise accordingly

Response: We thank the reviewer for this revision. Changed as suggested. This sentence will be revised in the manuscript as "It is assumed that the variation of the real bias within each hour is negligible" (P8L8)

10. P7L25: "so initial conditions of KF are. . ." I don't believe that the exact numbers for Q, S etc are a result of the previously stated assumption. Revise accordingly.

Response: We thank the reviewer for pointing this out. This sentence is revised as "the initial estimator for mean field radar rainfall logarithmic bias and it's error variance are assumed to equal their update values which are respectively the $BIAS_{KF}(0)$ and $P_{KF}(0)$.

11. Be consistent with the reporting of equations. Some are in text instead of being numbered as others. Also in P8,L9, you should write log[If(D)] instead of log(I). Revise also the sentence stating "β here accounting for nearly 50% occurrence probability. . ..". It is the intercept, not the exponent that relates to the probability according to the frequentist approach you used.

Response: Totally agree! log[If(D)] is written instead of log(I) in P8,L9. The modification is made as "where $\alpha_{50}$ , β is the fitted intercept and slope, respectively".(P8L23)

12. P9,L1 and elsewhere: use "scenarios" instead of "scene"

Response: Totally agree! The "scene" is replaced with "scenarios" in the revision.

13. 13. Equations 14 and 15 have the same formula. Please revise

Response: Equation 14 is normalized standard error (NSE), equation 15 is normalized mean bias (NMB).We checked those equations as suggested.

14. Define what do you mean by "linear ratio".

Response: The linear ratio is the slope estimated from linear regression of radar rainfall estimation and rain gauge observation, with the predefined intercept of zero. The linear ratio here is used to evaluate how much average ratio radar-based rainfall is to the observation of rain gauge. The linear ratio approximates to one, if radar-based rainfall estimation is consistent with rain gauge observation. We clarified this in the revision. (P10L3)

15. P10,L10-11: "The PDF estimations reveal that the number of positive difference δ(D) is more than number of negative difference". I am not sure what the point you are trying to make here is. Also, if you think that the distribution of residuals is asymmetric, then you should not fit a Gaussian distribution. This affects also the frequentist approach you followed. Please revise/clarify this point.

Response: We thank the reviewer for the comment. Perhaps it was not clear in the writing. But we would like to note that although the distribution of residuals is not strictly symmetric for low probability density, the sole peak and high probability density are conform to Gaussian distribution. The related sentences were eliminated in the revision.

---

## Author Response (AR1)

This file mainly consists of three parts. The first one is the point-by-point response to the comment of three referees. The second one is a list of all relevant changes made in the manuscript. The third one is the revised manuscript, in which the relevant change was marked up with yellow.

**Response to Anonymous Referee #1**

**General comments**

The paper deals with debris-flow occurrence thresholds in terms of rainfall intensity-duration by using radar data in an earthquake-affected area (Sichuan, China). The paper addresses technical questions within the scope of NHESS and is conform to international standards; scientific methods are clearly outlined. Authors try to address the definition of the rainfall-field by the use of radar to better evaluate the relation between rainfall and debris-flow. But, as described in the technical comments, to overcome some objective limitations of the methodology used, further investigations would be needed to properly take into account the susceptibility of the territory. The title is clear; the abstract is pertinent, easy to understand and resumes well the contents of the paper. Mathematical formulas, symbols and abbreviations are correctly defined. Technical language is precise and understandable.

An I-D threshold curve is proposed on the basis of six rain events that caused 512 debris-flows between 2012 and 2014. Based on rain data estimates obtained from 2 radars, the authors use various techniques for estimating rainfall from reflectivity, by also considering the rainfall detected by rain gauges for comparative analysis and correction of the bias. As shown, the use of radar data provides a better estimate of the precipitations responsible of landslides triggering. In one case (cf. Table 2), the amount of rain measured by radar is 10 times greater than that detected by the rain gauge network (Event # 3). Before evaluating the I-D threshold, the authors describe some techniques to detect the best estimate of radar precipitation, identifying the Kalman's filtering application as the best tool to reduce radar-rain gauge bias. Using the frequentist approach, the authors compute the relation I = αD-β which binds intensity and duration of precipitation for three estimates of rainfall fields. Scatter plots are shown in fig. 8

Response: We sincerely thank the reviewer for giving so helpful suggestions. In this paper we try to evaluate the feasibility and limitation for radar-based regional I-D thresholds over earthquake affected area in China. The information on hydrology, lithology, land use, geomorphology over study area could enhance the understanding of debris flow initiation. Some information on underlying surface are provided in the following paragraph describing the response. However doing an in-depth delving into those topics is beyond the scope of this paper.

**Technical Comments**

The article is of interest as it performs a radar application on I-D threshold models, historically developed using only rain gauges. Although, I-D thresholds generally suffer from limitations that can affect practical applications. In fact, in case of convective rainfall events (responsible of most debris-flow and shallow-landslide events), the area affected by ground effects is usually small (normally few km2) and tends to reproduce a low I-D curve when only rain gauges are used. On the other hand, landslides occur in correspondence of highest rainfall intensities. Aiming at evaluating the precipitation field, authors try to solve the problem by also using radar data, and considering possible estimation errors. Another intrinsic lack of the

method lies in the determination of the duration of the event, and the simplification of the average intensity or accumulated precipitation for the whole interval. Analyses of uniform rainfall distribution inside the intervals excludes soil characteristics that heavily affect the development of gravitational movements. In the case of stratiform rain events, with uniform intensity, landslides are more frequent in permeable areas; in case of convective rainfalls, landslides are more easily triggered in areas with low permeability. An additional limit of the model is the absence of discriminant analysis of susceptibility to shallow landslides in the study area. The adopted method for determining the threshold I-D, in fact, ignores lithology, geomorphology (e.g. slope), land use, and soil coverage. Consequently, it tends to lower the threshold as is results mostly controlled by the most fragile portions of the territory. Looking at the threshold I-D of Figure 8 for the same duration of precipitation, the corresponding rain intensity that triggers debris-flow varies with one order of magnitude. The paper, assessing thresholds I-D using radar rainfall field, contributes to improve the model by reducing limitations due to the use of only rain gauges, especially in the case of convective phenomena. Among possible improvements to the method, multiple thresholds (e.g. for class of slope, lithology, etc.) could be considered in addition to accurate rainfall radar estimation.

Response: We thank the reviewer for the valuable and constructive comment. We totally agree with the reviewer that the detailed spatial information on the hydrological, lithological, morphological, and soil characteristics could help us better understand the physical initiation of shallow landslides. In the revision, we have added the information about lithology, land use and morphology. In addition, potential debris flow watersheds are retrieved. We have updated the manuscript to include these points (P3L27). However, we want to note that it is challenging to get the high resolution high quality data continuously for this area, such as soil water status and soil drainage. Therefore, the empirical model is adopted in this study. The manuscript was supplemented with the following contents:

"The geological structure of the study area show a northeast to southwest orientation. The rocks over this region are mainly comprised of volcanic rocks, mixed sedimentary rocks, siliciclastic sedimentary rocks, carbonate sedimentary rocks, acid plutonic rocks, intermediate colcanic rocks, intermediate plutonic rocks, unconsolidated sediments, metamorphic rocks, basic Plutonic Rocks, and pyroclastic rocks. Figure 1a shows the lithological map. Quaternary deposits were distributed in the form of river terraces and alluvial fans. Owing to frequent tectonic activities, most of the gully are steeply sloped over this area. The main land use types in this region are mixed forest, cropland, and grassland, as shown in Figure 1b."(P.3 L.27)

[Figure]

Figure 1. Lithology map (a) and land use map (b) for the study area

"Potential DF watersheds over study area is extracted from morphological variables, using the logistic regression method. Berenguer et al.(2015) simplified the geomorphological variables, as the watersheds maximum height($h_{max}$), mean slope($s_{mean}$), mean aspect($\theta_{mean}$) and melton ratio(MR) are the variables with the smallest overlapping areas for assessing the susceptibility of the watersheds. The $h_{max}$, $s_{mean}$, $\theta_{mean}$ and MR were retrieved from DEM data. Combined with the DF occurrence over this area in the three years, the potential susceptibility map was calculated with logarithm regression method as shown in the figure 2c. "(P.4,L.2)

[Figure]

Figure 2. Morphology and potential DF watersheds map over study area.(a) slope, (b) aspect, (c) potential DF watersheds(gray polygon) with DF observation(blue circle)

"The identification results show that there are 673 potential debris flow watersheds in this region. 519 debris flows triggered by six rainfall events (point data) are shown in Figure 2c. In total, 98.6 % of the identified DF watersheds are located in the potential DF watersheds. "

Concerning the empirical relation of rainfall and debris flow, we adopted the widely used model described in (Guzzetti et al., 2007) , as D here is the duration from the beginning of the rainfall to the occurrence of the debris flow (h), and I is the mean rainfall intensity in the period of D (mm h$^{-1}$). The duration and intensity are determined by an interval of at least 24 hours, rain rates of less than 0.1 mm h$^{-1}$, or correspondingly radar reflectivity of less than 10 dBz to separate two consecutive rainfall events. We also note that there are various definition of rainfall thresholds for the initiation of debris flow in lots of literatures. Based on reviewer's constructive comment, we also would like go further to apply those definitions and investigate its performance in the following study by using the radar data.

In a word, we sincerely thank the reviewer for giving us so insightful comments and constructive suggestions.

Reference:

Guzzetti, F., Peruccacci, S., Rossi, M., and Stark, C. P.: Rainfall thresholds for the initiation of landslides in central and southern Europe, Meteorology and atmospheric physics, 98, 239-267, 2007.

M. Berenguer, D. Sempere-Torres, and M. Hürlimann.: Debris-flow forecasting at regional scale by combining susceptibility mapping and radar rainfall.Nat. Hazards Earth Syst. Sci., 15, 587–602,2015.

General Comments: This manuscript investigates the debris flow occurrence using rainfall Intensity-Duration (I-D) information in the earthquake-affected areas in Sichuan, China. Two S-band Doppler radars are used to estimate rainfall in the study domain characterized by complex terrain. Therein, traditional Z-R relations are adopted with additional attention paid to vertical profile of reflectivity (VPR) correction and Kalman filter based bias correction. The I-D curves are then calculated using a frequentist approach with the radar derived rainfall products.

Response: We sincerely thank the reviewer for valuable and constructive suggestions. The followings are our responses to the reviewer.

Overall, this topic well fits the scope of NHESS. The methodology used in the paper is scientifically sound. The study shows several valuable scientific results, which can possibly be used for guidance of debris flow warnings. Nevertheless, some minor changes will improve paper clarity. Specific comments are listed below:

1.  The writing needs to be improved. Please proofread the manuscript before submitting the revised version. Specific examples of mistakes are presented below: 1.1. Page 2, Line 15: highly rely-> highly relies 1.2. Page 3, Line 1: have to be -> has to be 1.3. Page 25, caption of Figure 7: event->events 1.4. Page 26, caption of Figure 8: possibility->probability

    Response: We thank the reviewer for the careful revision. Those were changed as suggested.

2.  Reference formatting should be more consistent throughout the manuscript. There are numerous references that are not cited in correct form. For examples, Qiang W. et al., 2015; Xiaojun G. et al 2016; Michele C. et al., 2015; Francesco M. et al., 2014; and many others.

    Additional Related Reference: Willie, D., H. Chen, and V. Chandrasekar, et al., 2017: Evaluation of Multisensor Quantitative Precipitation Estimation in Russian River Basin. Journal of Hydrologic Engineering, 22(5), E5016002, doi: 10.1061/(ASCE)HE.1943- 5584.0001422.

    Response: We thank the reviewer for pointing this out. The references are reformatted in accordance with Endnote template of Copernicus Publications. This reference suggested by reviewer also emphasized the combination of using radar and rain gauges to provide accurate rainfall estimates in complex terrain. It shows the necessity of radar-based rainfall estimation for improving warnings of future precipitation and situational awareness. We referred this article in the revision. (P.3,L4)

3.  Page 2, Line 14: Early Warning System Using acronym would be enough since it had been mentioned earlier.

    Response:Totally Agree! The acronym is used as suggested.

4.  Page 4, Line 5: performance specifications->system specifications

    Response: We thank the reviewer for pointing this out. Totally agree. Changed as suggested.

5.  Page 5, Lines 14-16: The authors are using different tilt data trying to obtain rainfall estimates at the same vertical level. What is the purpose of doing this? Getting rainfall closer to surface might be more useful (provided that lowest level data is not blocked).

Response: We thank the reviewer for this comment. We agree the idea of getting rainfall closer to surface with the reviewer. However, in our study, the hybrid mode using multiple elevation is not only to get the rainfall closer to surface for complex terrain, but also used to align the radar data from same altitude for the flat area where dense rain gauge are deployed. This manner here is expected to decrease the uncertainty of altitude factor (earth curvature) for correction using rain gauge. For sure, it might be more useful to get rainfall closer to surface, if radar bias and rainfall microphysics are well understood.

6. Page 6, Line 23: The rationale of using these two Z-R relations is insufficient. It is well known that Z-R is greatly dependent on local rainfall microphysics. A local DSD- based variability analysis would be helpful.

Response: We thanks the reviewer for this constructive suggestion. The DSD observation and microphysics retrieval has been proven beneficial to improve the accuracy of radar-based rainfall estimate, however, the DSD observations are rarely scarce for study area. It's believed that building the composite rainfall observing network will improve the QPE accuracy, furthermore enhance the performance of EWS.

7. Page 15, Table 1: doppler->Doppler

Response: Totally Agree! Changed as suggested!

8. Page 15, Table 1: The pulse width seems not matching with the range resolution. Please clarify.

Response: We thank the reviewer for this detailed check. The original transmitted pulse is 1μs, while the final filed variables are averaged within two range gates, which is pre-defined operational mode and cannot be changed by data user. Therefore, the range resolution is decreased from 150m to 300m. To avoid the misunderstanding for reader, the item of transmitted pulse will be remove from the table 1.

The manuscript presents an analysis of rainfall intensity-duration (I-D) thresholds used for the identification of debris-flow occurrence. Estimation of gauge and radar-based I- D threshold is carried out and compared. The work in this manuscript is very similar to the one carried out by Marra et al. 2014 thus from a methodological point of view there is no significant novelty. However, the authors carry out the analysis in a completely different region with different hydroclimatic characteristics and as such I consider the results to be complementary to what we already know from past studies. Therefore, I consider overall that the results reported in this work add to our knowledge and further highlight the significance of using remote sensing observations for the estimation of debris flow triggering rainfall. I am including below a list of comments/suggestions that can hopefully help the authors to improve their manuscript.

We sincerely thank the reviewer for giving us so valuable and constructive suggestions. The followings are our responses to the reviewer.

1. Page 4, L11-25: This last paragraph should be placed in a different section (not the study area and data). You could have a dedicated section to discuss event characteristics. Also in that same paragraph, you mention info that relates to methodology (e.g. identification of individual rainfall event) that should be placed in the methodology section.

   Response: We thank the reviewer for this helpful suggestion. The contents of this paragraph are adjusted according to this comment. Those contents related to event characteristic and identification of individual rainfall event were moved to the section 4 in the revision.(P8L22-P9L7)

2. Provide a more detailed analysis of the comparison between radar rainfall estimates at DF (debris flow location) and closest-gauge estimates. For example, a graph showing relative error (y-axis) vs distance (between closest gauge and DF) would be informative. Using different colors per event (on such a graph) would also provide some more info. Lastly, it would be interesting to show that for both rainfall intensity and duration, since you are reporting differences in duration as well. Differences in duration, although important for building I-D thresholds, are not frequently explored. I believe adding some more info on this would strengthen the overall analysis.

   Response: We thank the reviewer for the valuable and instructive suggestion. Based on this constructive comment, we added a new subsection 4.3 of 'Impact of rainfall spatial variation on Intensity and Duration'

   The accumulated rainfall, duration and rainfall intensity identified from the nearest rain gauge probably are different from the realities occurred at the debris flow location, since the rainfall varies in space especially for convective precipitation with sharp variation in short distance. The observed rainfall differences rely on the distance from the nearest rain gauge to the debris location and could be considered as spatial errors. To this end, relative errors of the accumulated rainfall, duration, and rainfall intensity versus distance are calculated from the comparisons with the radar-based estimate at the location of debris flow. The metrics for evaluating relative error versus distance are defined in the following table (Table 5 in the revision). There are also some predefined conditions for the comparison of relative errors versus distance: (1) The radar rainfall estimation used for comparison are all from scenario III. (2) The radar rainfall estimation and duration identification at the debris flow location are considered as the true referred value. (3) The maximum distance from debris

flow location to the nearest rain gauge is predefined within 10 km and the distance resolution is set equal to those two CINRADs' range resolution of 300 meters. (4) In order to assess the rainfall spatial variation using multi-sensor, the radar-based estimate at the co-location of the nearest rain gauge, as well as rain gauge observation, is also compared with the radar-based estimate at the location of debris flow.

Table 1. The metric for assessing the relative errors of the accumulated rainfall, duration and rainfall intensity versus distance

| Factors | Rain gauge observation nearest to DF location versus Radar estimate at DF location | Radar estimate at the co-location of rain gauge versus Radar estimate at DF location |
|---|---|---|
| Accumulated Rainfall Relative Error (ARRE) | $ARRE_g(s) = \dfrac{\sum_{i=1}^{N(s)}\left|R_{df}(i) - R_g(i)\right|}{\sum_{i=1}^{N(s)} R_{df}(i)} \times 100\%$ | $ARRE_r(s) = \dfrac{\sum_{i=1}^{N(s)}\left|R_{df}(i) - R_r(i)\right|}{\sum_{i=1}^{N(s)} R_{df}(i)} \times 100\%$ |
| Duration Relative Error (DRE) | $DRE_g(s) = \dfrac{\sum_{i=1}^{N(s)}\left|D_{df}(i) - D_g(i)\right|}{\sum_{i=1}^{N(s)} D_{df}(i)} \times 100\%$ | $DRE_r(s) = \dfrac{\sum_{i=1}^{N(s)}\left|D_{df}(i) - D_r(i)\right|}{\sum_{i=1}^{N(s)} D_{df}(i)} \times 100\%$ |
| Rainfall Intensity Relative Error (RIRE) | $RIRE_g(s) = \dfrac{\sum_{i=1}^{N(s)}\left|I_{df}(i) - I_g(i)\right|}{\sum_{i=1}^{N(s)} I_{df}(i)} \times 100\%$ | $RIRE_r(s) = \dfrac{\sum_{i=1}^{N(s)}\left|I_{df}(i) - I_r(i)\right|}{\sum_{i=1}^{N(s)} I_{df}(i)} \times 100\%$ |

Note. $R$ represents accumulated rainfall for debris flow event, $D$ represents duration for rainfall event, $I$ represents the mean intensity for rainfall event. The variables with subscript df, g and r respectively represent the observation from radar at debris flow location, rain gauge nearest to debris flow location, and radar at the co-location of the nearest rain gauge. s represents the distance between the nearest rain gauge location and debris flow location with the range resolution of 300m. $N(s)$ represent the number of rain gauge observation for debris flow at the distance of $s$.

The metrics of Accumulated Rainfall Relative Error (ARRE), Duration Relative Error (DRE) and Rainfall Intensity Relative Error ( RIRE) are calculated for the nearest rain gauge and radar estimate at the co-location respectively. The results of ARRE, DRE and RIRE versus distance are drawn in the following Figure (Figure 13 in the revision). The main findings from the evaluation results are summarized as follows:

(1) The results of ARRE, DRE and RIRE all have an enlarging tendency along with the increasing distance. The maximum ARRE, DRE and RIRE for rain gauge observation is 42.2%, 41.67% and 55.88%, respectively. The maximum ARRE, DRE and RIRE for radar-based estimate at the co-location of the nearest rain gauge is 43.33%, 41% and 45.2%, respectively.

(2) Nonlinear regression is applied for ARRE, DRE and RIRE versus distance to investigate the average tendency, as shown in the following figure (Figure 13 in the revision). The regression curves of ARRE and DRE for rain gauge and

radar are approximately similar within 10km and 4km, respectively, indicating radar estimate and rain gauge observation have the similar rainfall spatial variation impacts.

[Figure]

Figure 1. Scatterplot of relative errors versus distance. Blue circle dot represent relative error between radar estimate at debris flow location and rain gauge observation closest to debris flow location. Red asterisk represent relative error between radar estimate at debris flow location and radar estimate at the position of closet rain gauge.(a) Accumulated Rainfall Relative Error, (b) Duration Relative Error, (c) Rainfall Intensity Relative Error.

We clarified those in the new section 4.3 in the revision.(P11L18-P12L7)

We were also inspired by the reviewer's excellent suggestion of using different colors per event (on such a graph). As far as the events in this study are concerned, we found there were not sufficient samples to estimate the statistical relation between relative error and distance for separate event 1, 2, 4 and 6, which would lead to incomparable for those events. We decided not to compare relative error versus distance for individual event. However, we were deeply impressed by this suggestion, and would collect more eligible events for further study.

3. Provide also quantification metrics for changes in I-D parameters (α and β).

Response: We thank the reviewer for this suggestion. We further take the α and β estimated from Scenario III as the right referred value, and calculate the relative error of α and β for each scenario, as shown in the following table (table 6 in the revision). The relative error of α for Scenario I, II, and III is -24.5%, -13.8% and 0%, respectively. The relative error of β for Scenario I, II, and III is -28.8%, -17.3% and 0%, respectively. It is indicated that improving the accuracy of rainfall estimate is able to decrease the relative errors of α and β, Considering rainfall spatial variation, the relative error of α for the nearest gauge observation and radar-based estimate at the co-location is -49.5% and -42.6%, respectively. The relative error of β for the nearest gauge observation and radar-based estimate at the co-location is -19.5% and -21.2%, respectively. The relative error of $\alpha$ is remarkably larger than the one derived from radar-based estimate at the debris flow location,

however, the differences of $\alpha$ and $\beta$ for rain gauges and radar-based estimate at the co-location are not significant. We clarified those in the section 4.3 of revision.(P12L8-17)

Table 2. Parameters of the identified ID thresholds and relative errors

| | $\alpha$ | $\dfrac{\alpha - \alpha_{S3}}{\alpha_{S3}} \times 100\%$ | $\beta$ | $\dfrac{\beta - \beta_{S3}}{\beta_{S3}} \times 100\%$ |
|---|---|---|---|---|
| Scenario I | 7.62 | -24.5 | 0.67 | 28.8 |
| Scenario II | 8.7 | -13.8 | 0.43 | -17.3 |
| Scenario III | 10.1 | 0.0 | 0.52 | 0.0 |
| Rain gauges | 5.1 | -49.5 | 0.42 | -19.2 |
| Radar estimate at the co-location of the nearest rain gauge | 5.8 | -42.6 | 0.41 | -21.2 |

Note. $\alpha_{S3}$, $\beta_{S3}$ here equals to $\alpha$, $\beta$ estimated from Scenario III, respectively.

4.  A professional or native English speaker needs to carefully edit the manuscript for grammatical errors and inappropriate wording (e.g. p10L1 "effectivity" p10L7 "induce" etc)

    Response: We thank the reviewer for this good suggestion. The whole manuscript were read and revised by native English speaker.

5.  P5L3: "ensure the rainfall estimation accuracy" is quite a strong statement. Please revise.

    Response: We thank the reviewer for pointing this out. Totally agree! The aim of processing is to improve the rainfall estimation accuracy, the sentence is revised as "improve the rainfall estimation accuracy".(P4L31)

6.  P5,L27: Define VIL

    Response: We thank the reviewer for pointing this out. The definition of Vertically Integrated Liquid (VIL) is added in the manuscript as "To discriminate convection precipitation from stratiform based on the composite reflectivity>50dBz or VIL >6.5 kg/m2, where VIL is acronym of Vertically Integrated Liquid water content and it is an estimate of the total mass of precipitation in the clouds. (Amburn and Wolf, 1997)" (P5L24)

7.  P5,L31-32: "It can be seen that . . .rely on temperature, air dynamic . . .". I don't think that these can be seen from Figure 4 alone. Please revise.

    Respone: We thank the reviewer for pointing this out. We clarified this in the revision as" Impacted by the temperature, air dynamic, particle size and phase are changed along the vertical falling. Figure 5 shows vertical profile of reflectivity varied approximately as three piecewise linear sections".(P5L29)

8.  Do you have any justification for the choice of 1.5km as the height threshold for separating the two regions?

    Response: The reason of separating two region is the vertical variation of rainfall rate profile, especially for convective rainfall. Normally, the low scanning elevation PPI is used to estimate ground rainfall rate. Considering the limitation of scanning elevation, station height, and earth curvature, the height from flat terrain is nearly 1.5 km if the radial range is

100 km (normally under maximum detection range) away from radar, even for the lowest elevation of 0.5°. Concerning complex terrain of study area, the elevation almost has to be uplifted to avoid beam blockage when radial distance is over 100 km. Therefore, 1.5 km is set as height threshold for this study to discriminate where is closer to ground and where is higher from ground. The following figure briefly illustrate the radar beam locating height along the radial distance.

[Figure]

Figure 2. Radar beam locating height along the radial distance

9. P7L24: "between each hour is tiny" perhaps should be "within each hour is negligible". Please check and revise accordingly

   Response: We thank the reviewer for this revision. Changed as suggested. This sentence will be revised in the manuscript as "It is assumed that the variation of the real bias within each hour is negligible" (P7L23)

10. P7L25: "so initial conditions of KF are. . ." I don't believe that the exact numbers for Q, S etc are a result of the previously stated assumption. Revise accordingly.

    Response: We thank the reviewer for pointing this out. This sentence is revised as "the initial estimator for mean field radar rainfall logarithmic bias and it's error variance are assumed to equal their update values which are respectively the $BIAS_{KF}(0)$ and $P_{KF}(0)$.(P7L23)

11. Be consistent with the reporting of equations. Some are in text instead of being numbered as others. Also in P8,L9, you should write log[If(D)] instead of log(I). Revise also the sentence stating "β here accounting for nearly 50% occurrence probability. . ..". It is the intercept, not the exponent that relates to the probability according to the frequentist approach you used.

    Response: Totally agree! log[If(D)] is written instead of log(I) in P8,L9. The modification is made as "where $\alpha_{50}$ , $\beta$ is the fitted intercept and slope, respectively".(P8L6)

12. P9,L1 and elsewhere: use "scenarios" instead of "scene"

Response: We thank the reviewer for pointing this out. Totally agree! Changed as suggested. All of "scene" are replaced with "scenarios" in the revision.

13. 13. Equations 14 and 15 have the same formula. Please revise

Response: Equation 14 is normalized standard error (NSE), equation 15 is normalized mean bias (NMB).We clarified those equations in the revision as suggested.

14. Define what do you mean by "linear ratio".

Response: The linear ratio is the slope estimated from linear regression of radar rainfall estimation and rain gauge observation, with the predefined intercept of zero. The linear ratio here is used to evaluate how much average ratio radar-based rainfall is to the observation of rain gauge. The linear ratio approximates to one, if radar-based rainfall estimation is consistent with rain gauge observation. We clarified this in the revision. (P9L20)

15. P10,L10-11: "The PDF estimations reveal that the number of positive difference δ(D) is more than number of negative difference". I am not sure what the point you are trying to make here is. Also, if you think that the distribution of residuals is asymmetric, then you should not fit a Gaussian distribution. This affects also the frequentist approach you followed. Please revise/clarify this point.

Response: We thank the reviewer for the comment. Perhaps it was not clear in the writing. But we would like to note that although the distribution of residuals is not strictly symmetric for low probability density, the sole peak and high probability density are conform to Gaussian distribution. The related sentences were eliminated in the revision.

The list of all relevant changes made in the manuscript

| NO. | Position in the revision | Changes |
|---|---|---|
| 1. | P2L14 | Revised:
Early Warning System-> EWS |
| 2. | P2L15 | Revised:
rely->relies |
| 3. | P3L1 | Revised:
have-> has |
| 4. | P3L4 | New added content:
Therefore, the combination of using radar and rain gauges to provide accurate rainfall estimates in complex terrain attracts increasingly more interest for improving warnings of future precipitation and situational awareness (Willie et al., 2017). |
| 5. | P3L18 | Revised:
Closest->nearest |
| 6. |  | All of the term 'scene' are replaced with 'scenario' |
| 7. | P4L18 | Revised:
performance-> system |
| 8. | P3L27 | New added content:

[revised manuscript text omitted]

---

## Author Response (AR2)

This file mainly consists of three parts. The first part is the point-by-point response to the comment of the referee. The second part is a  list of all relevant changes made in the manuscript. The third part  is the revised manuscript, in which the relevant changes were marked up with yellow.

**Response to Anonymous Referee**

**General comments**

I would like to congratulate the authors for carrying out substantial work and submitting a significantly improved manuscript. I believe that the new section 4.4 complements nicely the work but it can further improve by clarifying/revising some statements. I provide some examples below:

> Response: We thank the reviewer for the kind words. Once again, we truly appreciate the time and efforts the reviewer spent in reviewing our manuscript. It is very encouraging to know the reviewer is satisfied with our revisions. We have further revised this manuscript based on the minor comments from the reviewer.

1. When comparing rainfall estimates at different locations, you should refer to "change" rather than "error". Reference to "error" can be considered correct when comparing at the same location and you have already assumed that an estimate (e.g. scenario III) corresponds to the "truth", I am fine with this.

> Response: We thank the reviewer for pointing this out and completely agree with reviewer. We tried to improve the radar rainfall estimate with many methods, and the accuracy improvement of rainfall estimate was demonstrated in the scenario III , however, the residuals still inevitably existed. To this end, for comparing rainfall estimates at different locations, we replaced the term "error" with "change" in the subsection 4.4 of the revision as the reviewer suggested.

2. P12,L9: "indicating....have the similar rainfall spatial variation impact", please rephrase/improve this statement. Alternatively, you can state that this shows that the observed difference as a function of distance is dominated by the natural spatial variability and any potential impacts from differences in rainfall estimates coming from different sensors are secondary (in the accumulation case) or negligible (for duration which is basically sensitive to detection alone and not estimation of magnitude).

> Response: We thank the reviewer for this valuable comment. Changed as suggested. We clarified this in the revision as "indicating the observed difference as a function of distance is dominated by the natural spatial variability, and the potential impact from differences in rainfall estimates coming from different sensors is secondary, especially for estimating duration."(P12L9)

Please try to elaborate a bit on the above and better "polish" your statements/findings in this section.

> Response: Once again, we would like to express our sincere thanks to the reviewer. We believe this manuscript has been significantly improved with the help of the reviewer's valuable comments. Thanks to the reviewer!

The list of all relevant changes made in the manuscript

| NO. | Position in the manuscript | Revision (before -> after) |
|---|---|---|
| 1. | P11L21 | Spatial errors -> rainfall spatial change |
| 2. | P11L21 | Relative errors -> relative changes |
| 3. | P11L23 | Relative error -> relative change |
| 4. | P11L24 | Relative error -> relative change |
| 5. | P11L26 | The true referred value -> the referred value |
| 6. | P11L31 | Accumulated Rainfall Relative Error (ARRE) ->Accumulated Rainfall Relative Change (ARRC) |
| 7. | P11L31 | Duration Relative Error (DRE) -> Duration Relative Change (DRC) |
| 8. | P11L32 | Rainfall Intensity Relative Error ( RIRE) -> Rainfall Intensity Relative Change ( RIRC) |
| 9. | P12L1 | ARRE, DRE and RIRE -> ARRC, DRC and RIRC |
| 10. | P12L3 | ARRE, DRE and RIRE -> ARRC, DRC and RIRC |
| 11. | P12L4 | ARRE, DRE and RIRE -> ARRC, DRC and RIRC |
| 12. | P12L5 | ARRE, DRE and RIRE -> ARRC, DRC and RIRC |
| 13. | P12L6 | ARRE, DRE and RIRE -> ARRC, DRC and RIRC |
| 14. | P12L7 | ARRE and DRE -> ARRC and RIRC |
| 15. | P12L9 | indicating the observed difference as a function of distance is dominated by the natural spatial variability, and the potential impact from differences in rainfall estimates coming from different sensors is secondary , especially for estimating duration. |
| 16. | P12L14 | Relative error -> relative change |
| 17. | P12L15 | Relative error -> relative change |
| 18. | P12L16 | Relative error -> relative change |
| 19. | P12L17 | Relative error -> relative change |
| 20. | P12L18 | Relative error -> relative change |
| 21. | P12L19 | Relative error -> relative change |
| 22. | P12L20 | Relative error -> relative change |

[revised manuscript text omitted]